# MECAT: A Multi-Experts Constructed Benchmark for Fine-Grained Audio Understanding Tasks

**Yadong Niu** [* 1]  **Tianzi Wang** [* 1 2]  **Heinrich Dinkel** [1]  **Xingwei Sun** [1]  **Jiahao Zhou** [1]  **Gang Li** [1]  **Jizhong Liu** [1]
**Xunying Liu** [2]  **Jian Luan** [1]

## Abstract

While large audio-language models have advanced open-ended audio understanding, they still fall short of nuanced human-level comprehension. This gap persists largely because current benchmarks, limited by data annotations and evaluation metrics, fail to reliably distinguish between generic and highly detailed model outputs. To this end, this work introduces MECAT, a **M**ulti-**E**xpert **C**onstructed Benchmark for Fine-Grained **A**udio Understanding **T**asks. Generated via a pipeline that integrates analysis from specialized expert models with Chain-of-Thought large language model reasoning, MECAT provides multi-perspective, fine-grained captions and open-set question-answering pairs. The benchmark is complemented by a novel metric: DATE (**D**iscriminative-**E**nhanced **A**udio **T**ext **E**valuation). This metric penalizes generic terms and rewards detailed descriptions by combining single-sample semantic similarity with cross-sample discriminability. A comprehensive evaluation of state-of-the-art audio models is also presented, providing new insights into their current capabilities and limitations. The code and data are publicly available at `https://github.com/xiaomi-research/mecat`.

## 1. Introduction

The human auditory system is highly effective at processing complex acoustic scenes. It can distinguish subtle variations in sound, such as telling a dog's playful bark from a defensive growl (Plack, 2023), and isolate target speech from noisy backgrounds, an ability known as the cocktail party effect. A central goal of machine hearing is to replicate this auditory intelligence to interpret raw audio signals as semantically rich perception (Lyon, 2017). Early works in machine hearing focused on closed-ended tasks such as sound event classification and automatic speech recognition. Large language models (LLM) have spurred the development of large audio-language models (LALMs), which have driven a shift towards more general open-ended tasks like audio captioning and audio question answering (Chu et al., 2023; Du et al., 2023; Hu et al., 2024; Shu et al., 2023; Wang et al., 2023; Tang et al., 2024; Rubenstein et al., 2023; Chen et al., 2023; Huang et al., 2024).

Despite these advances, current LALMs still fall short of achieving the comprehensive understanding that characterizes human hearing (Sakshi et al., 2025). This work argues that despite ongoing improvements in model architectures and data, a crucial and often-overlooked bottleneck is the existing evaluation benchmark.

The first challenge lies with data annotations, which suffers from several limitations. To begin with, the annotations in current benchmarks are often overly simplistic, consisting of event-level captions that lack detail (Mei et al., 2024; Kim et al., 2019; Drossos et al., 2020) or question-answering tasks confined to close-ended formats (Lipping et al., 2022; Wang et al., 2025; Sakshi et al., 2025). Furthermore, they typically adopt a monolithic perspective, providing a single, global description that fails to account for the selective nature of human hearing. Compounding these issues is a "one-to-many" data redundancy problem, where the same audio clips, often from AudioSet (Gemmeke et al., 2017), are reused across multiple benchmarks, limiting the assessment of model generalization.

The second challenge is rooted in evaluation metrics. Traditional lexical-matching metrics, on the one hand, penalizes semantically correct but lexically different descriptions. Embedding-based metrics, on the other hand, better align with human perception, they often fail to distinguish between generic, vague captions and highly detailed, accurate ones. Even the more recent LLM-as-judge method, while

---
[*]Equal contribution  [1]MiLM Plus, Xiaomi Inc, Beijing, China
[2]The Chinese University of Hong Kong, Hong Kong, China.
Correspondence to: Yadong Niu <niuyadong@xiaomi.com>,
Heinrich Dinkel <dinkelheinrich@xiaomi.com>, Tianzi Wang
<twang@se.cuhk.edu.hk>.

*Proceedings of the 43rd International Conference on Machine Learning*, Seoul, South Korea. PMLR 306, 2026. Copyright 2026 by the author(s).

demonstrating strong discriminative ability, is often hindered by practical constraints such as high costs and slow inference speeds, as well as its high dependency on model selection and prompt design.

Thus, current benchmarks inadequately evaluate audio understanding, as they often reward generic captions (e.g., A dog is barking and people are talking) for distinct scenarios (e.g., an excited bark in a park vs. a defensive bark during an argument). This limits their ability to differentiate between models with true perceptual accuracy and those producing vague outputs.

To this end, we introduce MECAT, a Multi-Expert Constructed Benchmark for Fine-Grained Audio Understanding Tasks. By integrating analysis from a series of specialized audio-related experts models, including content-specific models (e.g., for speech, music, and sound events) and content-unrelated models (e.g., for audio quality, reverberation and intensity), followed by Chain-of-Thought (CoT) enhanced LLM reasoning (Guo et al., 2025), MECAT provide fine-grained captions alongside open-set question-answering pairs. The captions primarily focus on providing a comprehensive, multi-perspective description of the acoustic scene, while the QA pairs are designed to probe for specific details and higher-level contextual reasoning that descriptive tasks alone cannot fully assess. Furthermore, we introduce a novel metric *DATE* (Discriminative-Enhanced Audio Text Evaluation), which is designed to better quantify the detail and accuracy of model's response. It uniquely combines a weighted single-sample semantic similarity that penalizes generic terms while emphasizing discriminative phrases, and a cross-sample discriminability score that explicitly rewards the model's responses for exceeding general descriptions. This design enables DATE to robustly distinguish between superficial and context-rich model outputs.

**Conflict of Interest Disclosure.** Several authors are employed by Xiaomi Inc., the developer of the MiMo-Audio model evaluated in this paper.

## 2. Related works

### 2.1. Audio Captioning Benchmark

Audio captioning benchmarks have been pivotal in advancing audio understanding works (Wu et al., 2019; Kim et al., 2019; Drossos et al., 2020; Yuan et al., 2025; Manco et al., 2023; Liu et al., 2024a;b). Early dataset like Audio-Caps (Kim et al., 2019) and Clotho (Drossos et al., 2020) primarily relied on manual annotation, where human annotators provide one or more captions for each audio clip. While foundational, these benchmarks face a critical limitation: the coarse-grained nature of their annotations. During the annotation process, human annotators often produce generic, events-level descriptions rather than capturing the nuanced acoustic details of a scene. This results in a gold standard that lacks the specificity needed to evaluate fine-grained understanding.

While newer methods using LLMs for automatic labeling, such as in AutoACD (Sun et al., 2024) and LPMusicCaps (Doh & Nam, 2023), have improved scalability, they did not solve the granularity problem. Caption quality suffers from coarse input metadata like titles and tags, perpetuating generic descriptions.

### 2.2. Audio Question-Answering Benchmark

Audio Question Answering (QA) presents a more targeted evaluation of a model's audio understanding abilities (Lipping et al., 2022; Wang et al., 2025; Li et al., 2022; Sakshi et al., 2025; Ma et al., 2025). Datasets like ClothoAQA (Lipping et al., 2022) and MusicAVQA (Li et al., 2022) have been developed with manually crafted question-answer pairs. However, similar to captioning benchmarks, they suffer from limitations that hinder the assessment of detailed understanding.

The main issue is their reliance on close-ended answer formats designed for easier automatic scoring. For example, many questions in ClothoAQA are limited to "yes/no" answers (Lipping et al., 2022), while other benchmarks like MMAU (Sakshi et al., 2025) utilize a multiple-choice format. While convenient for evaluation, these formats prevent the assessment of a model's ability to generate detailed, descriptive answers and may encourage models to learn shallow pattern matching rather than deep understanding.

### 2.3. Evaluation Metrics for Audio Caption and QA

The evaluation of open-ended audio caption and QA is critically dependent on the choice of metric. However, existing metrics fail to adequately assess the fine-grained descriptive capabilities of modern generative models.

Traditional metrics, such as BLEU (Papineni et al., 2002), CIDEr (Vedantam et al., 2015), and SPICE (Anderson et al., 2016), operate by measuring lexical overlap with reference texts. This reliance on n-gram matching unfairly penalizes novel yet accurate descriptions that do not share the exact wording of the references.

To overcome the limitations of lexical matching, embedding similarity-based metrics were introduced. Approaches like FENSE (Zhou et al., 2022) were specifically designed for audio captioning. However, our experiments found that they still struggle to effectively distinguish between a generic, vague response and a highly detailed and accurate one. More recently, LLM-as-judge has been adopted for evaluating open-ended generation (Wang et al., 2025; Zheng et al., 2023). These methods show strong correlation with human

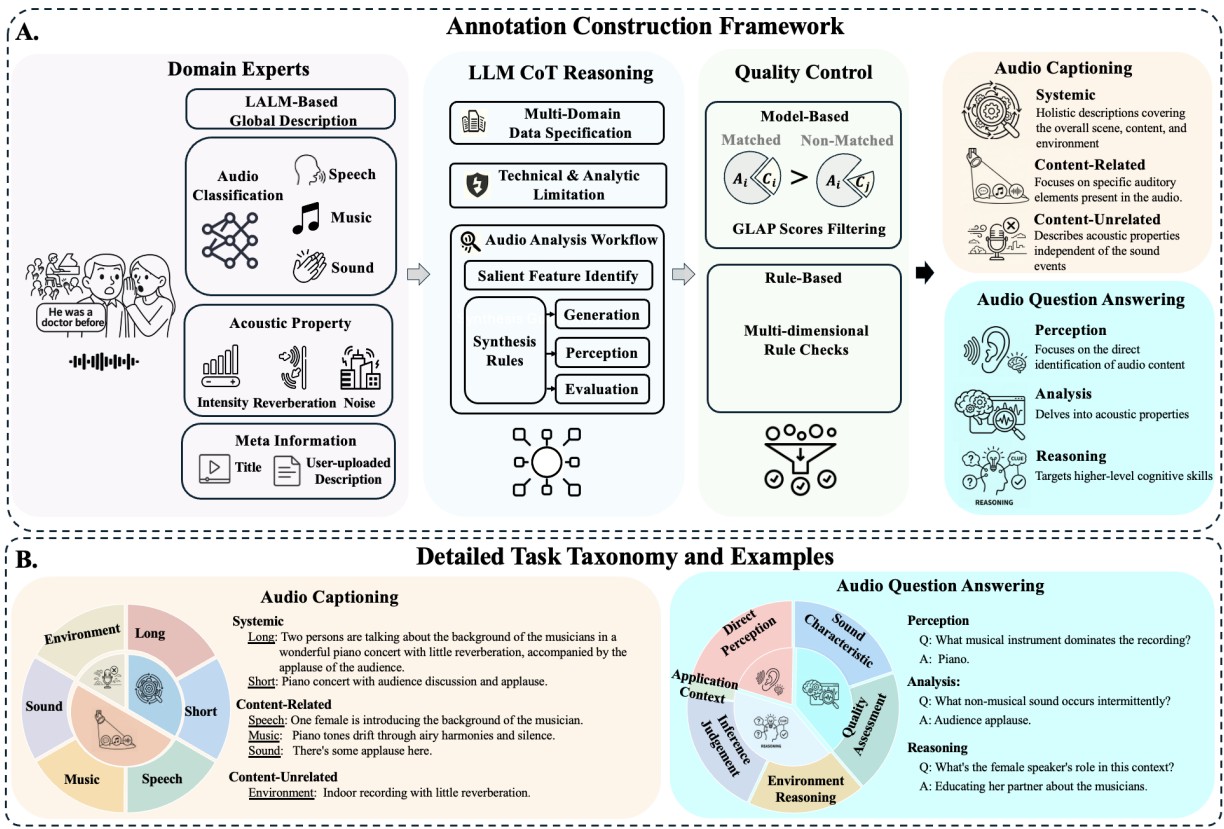

*Figure 1.* Overview of the MECAT Benchmark. (A) The proposed annotation construction pipeline. (B) Detailed task taxonomy and representative examples for Audio Captioning and Question Answering, showcasing the diversity of the dataset.

judgment and, as our experiments confirmed, possess a high sensitivity to response specificity. However, LLM-as-judge suffer from practical limitations, such as high computational costs, slow evaluation speeds, and a strong dependency on the choice of the LLM and the design of the prompt (Lee et al., 2024; Zheng et al., 2023).

## 3. MECAT Benchmark Overview

As illustrated in Figure 1, MECAT is a comprehensive benchmark for fine-grained audio understanding, distinguished by its unique data sources, broad domain coverage, and two core evaluation tasks: MECAT-Caption and MECAT-QA.

### 3.1. Dataset Description

To ensure data source novelty, MECAT is constructed from a carefully selected subset of ACAV100M (Lee et al., 2021). This approach contrasts with benchmarks, such as Audio-Caps (Kim et al., 2019), Clotho (Drossos et al., 2020), and WavCaps-QA (Wang et al., 2025), which predominantly draw from a limited pool of sources such as AudioSet (Gemmeke et al., 2017) and Clotho (Drossos et al., 2020) (see Table 1). The dataset comprises approximately 20,000 Cre-

ative Commons-licensed audio clips, each with a maximum duration of 10 seconds which is sufficient to contain one or a few salient acoustic events, while still allowing us to attach dense supervision for fine-grained, clip-local understanding.

Based on this unique data foundation, MECAT encompasses eight distinct audio domains designed to comprehensively represent real-world acoustic scenarios. These categories include four *Pure* domains: silence (000), speech (S00), sound events (00A), and music (0M0), as well as all four

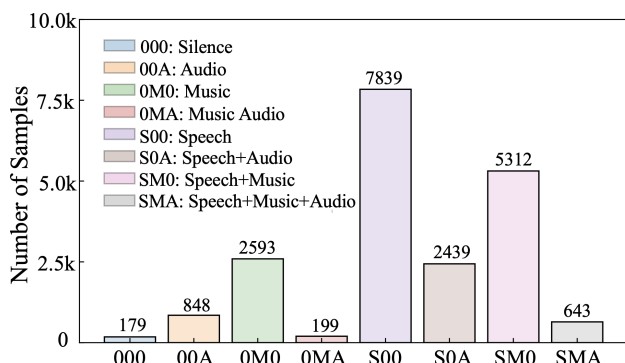

*Figure 2.* Distribution of audio samples across extended multi-domains in the MECAT, including speech, music, audio, combinations thereof, and silence.

*Table 1.* Comparison of MECAT with Recent General Sound Evaluation Benchmark Datasets. † MP-LLM: Multiple Experts Models and LLM; ‡ Multi-Domain: This includes speech, music and sound-events (⋄ denotes that domain were not elaborated in detail); § Extended Multi-Domain: This includes speech, music, sound-events, combinations thereof, and silence.

| Task | Labeling | Dataset | Test Size | Domain | Source |
|------|----------|---------|-----------|--------|--------|
| Caption | Manual | AudioCaps (Kim et al., 2019) | ∼1.0k | Multi-Domain[‡,⋄] | AudioSet |
| | | Clotho (Drossos et al., 2020) | ∼1.0k | Multi-Domain[‡,⋄] | Clotho |
| | | SongDescriber (Manco et al., 2023) | 0.7k | Music | MTG-Jamendo |
| | LLM | AudioCaps-Enhanced (Yuan et al., 2025) | 0.9k | Multi-Domain[‡,⋄] | AudioSet |
| | | AutoACD (Sun et al., 2024) | 1.0k | Multi-Domain[‡,⋄] | AudioSet |
| | | LPMusicCaps-MSD (Doh & Nam, 2023) | 35k | Music | Song Dataset |
| | | LPMusicCaps-MTT (Doh & Nam, 2023) | 5k | Music | MagnaTagATune |
| | MP-LLM[†] | MECAT-Caption (Ours) | 20k | Extended Multi-Domain[§] | ACAV100M |
| QA | Manual | ClothoAQA (Lipping et al., 2022) | 2k | Multi-Domain[‡,⋄] | Clotho |
| | | WavCaps-QA (Wang et al., 2025) | 0.3k | Multi-Domain[‡,⋄] | AudioSet and 2 others |
| | | MusicAVQA (Li et al., 2022) | 6k | Music | YouTube |
| | | Audiocaps-QA (Wang et al., 2025) | 0.3k | Multi-Domain[‡,⋄] | AudioSet |
| | | MMAU (Sakshi et al., 2025) | 10k | Multi-Domain[‡] | AudioSet and 12 others |
| | LLM | EvalSIFT (Pandey et al., 2025) | 30k | Speech | Open-source ASR |
| | MP-LLM[†] | MECAT-QA (Ours) | 20k | Extended Multi-Domain[§] | ACAV100M |

possible combinations of *Mixed* domains that reflect the complexity of natural auditory environments (SM0, S0A, 0MA, and SMA). This extended multi-domain coverage, with its distribution detailed in Figure 2, enables a nuanced evaluation of models on complex acoustic scenes, such as those that combine piano music with spoken discussion and audience applause.

### 3.2. Tasks Definition

As illustrated in Figure 1-B, the MECAT-Caption task delivers multi-perspective annotations for comprehensive evaluation. Each audio clip is annotated with a rich set of captions organized into three categories, which together comprise six distinct sub-categories. The first category, *Systemic Captions*, consists of two sub-categories: a concise short caption focused on primary audio content and a detailed long caption encompassing contextual details and event interactions. The second category, *Content-Specific Captions*, includes three sub-categories for the independent analysis of speech, music, and sound events. Crucially, to assess model performance across different levels of acoustic complexity, the evaluation for each content type is performed on corresponding pure domains (e.g., pure speech - S00) and all mixed domains. Notably, these captions also explicitly state when a corresponding domain is absent. The final category is a single *Content-Unrelated Caption* that focuses exclusively on acoustic characteristics like audio quality and reverberation. For each of these six sub-categories, three synonymous reference captions are provided, yielding a total of 18 reference captions per clip and creating a significantly richer vocabulary than existing datasets (see Appendix B for more details).

The final score $\text{Score}_{\text{Cap}}$ for the MECAT-Caption task is calculated as a weighted average of the three main categories:

$$\text{Score}_{\text{Cap}} = 0.4 \cdot S_{\text{Systemic}} + 0.4 \cdot S_{\text{Content-Specific}} + 0.2 \cdot S_{\text{Content-Unrelated}}. \quad (1)$$

where the category scores are themselves weighted sums of their sub-categories:

$$S_{\text{Systemic}} = 0.8 \cdot S_{\text{Long}} + 0.2 \cdot S_{\text{Short}}, \quad (2)$$

$$S_{\text{Content-Specific}} = 0.6 \cdot S_{\text{Speech}} + 0.3 \cdot S_{\text{Music}} + 0.1 \cdot S_{\text{Sound}}. \quad (3)$$

The score for each content type ($S_{\text{Speech}}$, $S_{\text{Music}}$, $S_{\text{Sound}}$) is calculated as the unweighted mean of its performance on the corresponding pure domains (e.g., S00, 0M0, 00A) and all mixed domains. All coefficients are set heuristically to reflect the relative importance of overall scene, content detail, and acoustic context. Within the Content-Specific category, the $0.6/0.3/0.1$ weights roughly follow the relative prevalence of Speech, Music, and Sound in ACAV100M. A sensitivity analysis confirms that model rankings remain highly stable across alternative weighting scenarios (Kendall's $\tau = 0.92$; see Section F.2.1).

Complementing the captioning task, MECAT-QA facilitates evaluation through targeted, probing questions. Each audio clip is paired with five question-answer pairs that span different cognitive skills, resulting in over 100,000 QA pairs in total. These pairs are organized into three cognitive categories. The first, *Perception*, focuses on the direct identification of audio content through its Direct Perception (DP) sub-category. The second, *Analysis*, delves into acoustic

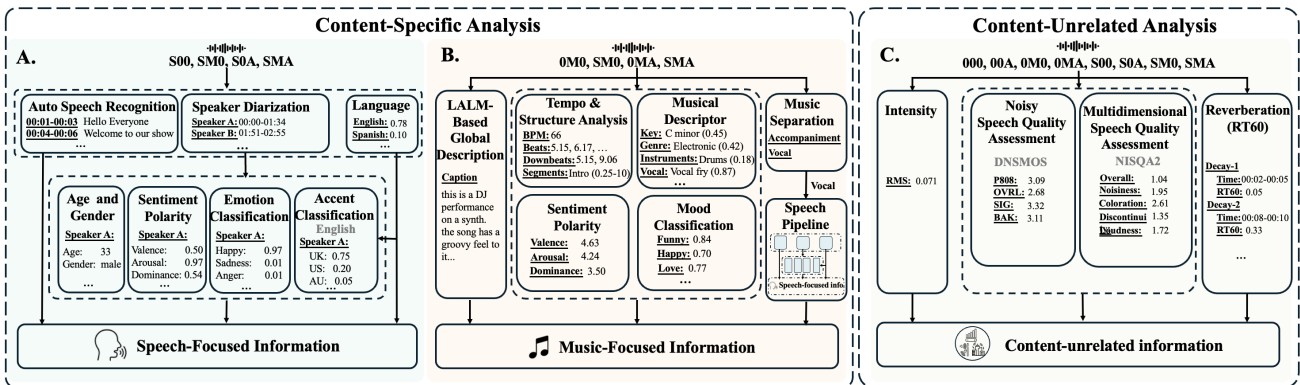

*Figure 3.* Domain Experts for Speech, Music, and Acoustic Properties.

properties via two sub-categories: Sound Characteristics (SC), for examining properties like pitch, and Quality Assessment (QAS), for evaluating technical fidelity. The final and most complex category, *Reasoning*, targets higher-level cognitive skills through three sub-categories: Environment Reasoning (ER), requiring acoustic scene inference; Inference & Judgement (IJ), involving logical deductions; and Application Context (AC), testing the understanding of practical scenarios.

The scoring for MECAT-QA is designed to ensure equal contribution from each cognitive skill. The overall score is the unweighted arithmetic mean of the scores from all six individual sub-categories:

$$\text{Score}_{\text{QA}} = (S_{\text{DP}} + S_{\text{SC}} + S_{\text{QAS}} + S_{\text{ER}} + S_{\text{IJ}} + S_{\text{AC}})/6. \quad (4)$$

## 4. Annotation Construction

This section details the MECAT annotation construction pipeline. As illustrated in Figure 1, the process starts with a audio classification stage identifying the domain of each audio clip. Based on the resulting domains, the clip is then processed by a series of specialized expert models.

The structured outputs from these experts are subsequently synthesized using LLM CoT reasoning to generate fine-grained captions and open-set QA pairs. The pipeline concludes with a rigorous quality control stage to ensure the reliability of all final annotations. The complete list of the used models is available in Appendix C.

### 4.1. Domain Experts

For each audio clip, we first use Audio Flamingo 2 (Ghosh et al., 2025) to generate a global, event-level summarization in natural language. Furthermore, we apply a series of domain expert models for more detailed analysis.

**Audio Classification** For each audio clip, we use CED-Base (Dinkel et al., 2024a) to predict AudioSet (Gemmeke et al., 2017) labels for every 2-second, non-overlapping

interval. This process results in a sequence of multi-label predictions for each clip. Based on the CED prediction, we categorize each clip into one of eight distinct domains: 000, 00A, 0M0, S00, SM0, 0MA, S0A, SMA, as detailed in Dataset Description Section.

**Speech-focused Analysis** For speech-domain clips (S00, S0A, SM0, SMA), we employ a speech-focused analysis pipeline(Figure 3-A). The pipeline consists of automatic speech recognition, language identification, and speaker diarization. Using the temporal boundaries from diarization, we extract each speaker's attributes, including gender, age, emotion, and English accent. The probabilities of these results are also utilized for subsequent LLM reasoning.

**Music-focused Analysis** For music-domain clips (0M0, SM0, 0MA, SMA), a music-focused analysis pipeline is employed (Figure 3-B). It consists of LALM-based global description of music content (Audio Flamingo 2 (Ghosh et al., 2025)), musical attribute analysis, and music separation. Musical attribute analysis provides a series of perceptual and technical attributes such as emotions and tempo. The music separation module isolates vocal tracks from the instrumental background, which are then routed to the speech analysis pipeline.

**Sound Events-focused Analysis** For audios in 00A, we directly utilize the events labels predicted by the CED-Base model during the audio classification stage.

**Acoustic Properties Analysis** To extract fundamental signal characteristics and assess the recording environment, we apply a universal acoustic property analysis pipeline to all audio clips (Figure 3-C). The analysis content includes signal intensity, speech quality assessment, and reverberation. Signal intensity is quantified via Root Mean Square (RMS). For audio quality, we conduct both DNSMOS (Reddy, 2021) and NISQA2 (Mittag et al., 2021) assessments to measure signal distortion, background noise, and perceptual quality. We also characterize the acoustic environment by estimating the reverberation time of the recording space.

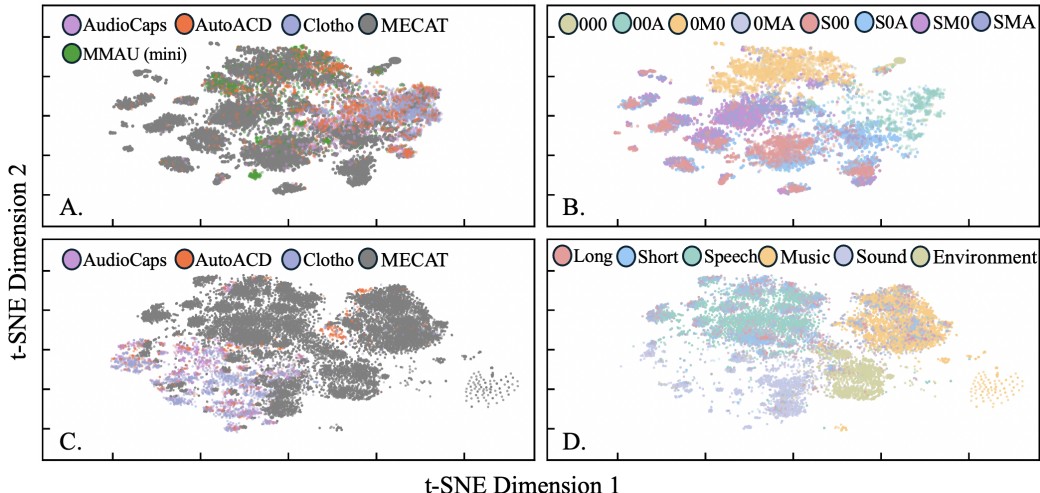

*Figure 4.* t-SNE plots of MECAT audio embeddings compared to other benchmarks (A), further clustered by domain (B). Caption embeddings are visualized in C and clustered by categories in D. Audio embeddings and captions embeddings are extracted from Dasheng-Base (Dinkel et al., 2024b) and Sentence-BERT (Reimers & Gurevych, 2019) respectively.

## 4.2. LLM CoT Reasoning

Our pipeline employs a Chain-of-Thought (CoT) guided LLM (Deepseek-R1: Guo et al.,2025) to synthesize a set of rich annotations. The model is instructed to reason over the outputs from all preceding analyses and the metadata. This reasoning process weighs evidence from various sources to resolve inconsistencies and identifying salient features. The final output consists of captions and corresponding QA pairs, where each item is annotated with a confidence level. The complete prompt is shown in Appendix D.

## 4.3. Quality Control

The model-based filtering use GLAP (Dinkel et al., 2026) to compute the cosine similarity between audio clip and its systemic long caption embeddings. A sample is kept only if the similarity of its correct audio-caption pair exceeds its average similarity with a set of 6 other randomly selected captions by an empirically set threshold of 6.

We further apply rule-based filtering including LLM confidence thresholding, domain consistency between audio classification and LLM output, and hallucination removal (Barański et al., 2025).

## 5. Metric Design

Existing evaluation metrics demonstrate significant limitations when evaluating fine-grained, detailed descriptions. To address this, we propose DATE, a metric built on Sentence-BERT (Reimers & Gurevych, 2019) that improves semantic assessment by combining single-sample semantic similarity and cross-sample discriminability score.

**Single-Sample Semantic Similarity** We apply embedding-level term frequency-inverse document frequency (TF-IDF) weighting to token embeddings from Sentence-BERT to emphasize tokens that are frequent within a single sample but rare across the dataset (details in Appendix E). The weighted embedding vector $\mathbf{v}_T$ for a given sentence $T$ is:

$$\mathbf{v}_T = \sum_{t \in T} \left( \text{TF}_{\text{emb}}(t, T) \cdot \text{IDF}_{\text{emb}}(t) \right) \cdot E(t), \quad (5)$$

where $t$ is a token in $T$. The term $\text{TF}_{\text{emb}}(t, T)$, $\text{IDF}_{\text{emb}}(t)$, and $E(t)$ are the frequency, inverse document frequency, and the Sentence-BERT embedding, respectively. The single-sample semantic similarity, $S_{\text{sim},i}$, is the cosine similarity between the weighted embeddings of the candidate and reference text:

$$S_{\text{sim},i} = (\mathbf{v}_{\text{cand}} \cdot \mathbf{v}_{\text{ref}}) / (\|\mathbf{v}_{\text{cand}}\| \|\mathbf{v}_{\text{ref}}\|). \quad (6)$$

**Cross-Sample Discriminability** An ideal description should be clearly distinguishable from descriptions of other audio samples. We construct a cross-sample similarity matrix, $\mathcal{M}$, where each element $M_{i,j}$ is the score between the reference description for audio $i$ and the candidate description for audio $j$. For each sample $i$, we rank the correctly matched score $M_{i,i}$ against all candidate scores $\{M_{i,j}\}_{j=1}^{N}$. Denoting this rank as $r_i$, the discriminability score is:

$$S_{\text{dis},i} = 1 - r_i / N. \quad (7)$$

This rewards candidates that rank highly for their correct reference, approaching 1 for top ranks and 0 for bottom ranks.

**DATE** To ensures a balanced evaluation for both descriptive accuracy and uniqueness, the DATE score for each sample $\text{DATE}_i$ is defined as the harmonic mean of its semantic similarity ($S_{\text{sim},i}$) and discriminability ($S_{\text{dis},i}$):

$$\text{DATE}_i = \frac{2 \cdot S_{\text{sim},i} \cdot S_{\text{dis},i}}{S_{\text{sim},i} + S_{\text{dis},i}} \in [0, 1]. \quad (8)$$

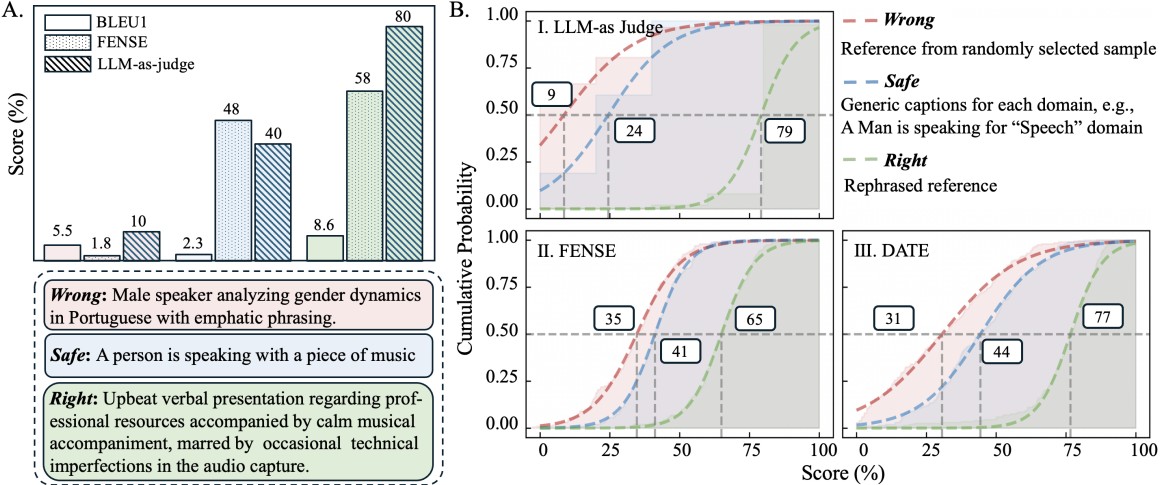

*Figure 5.* Metric Analysis. (A) Case study of existing metrics. Reference: "An animated woman's voice shares information about learning materials while melodic instruments play quietly underneath, with persistent low-quality artifacts in the recording." (B) Cumulative Distribution Functions (CDF) of LLM-as-judge, FENSE, and DATE on Caption (left) and QA (right). Larger distances between CDF curves indicate better discriminative ability of the metric.

The DATE score of a dataset is $\frac{1}{N} \sum_{i=1}^{N} \mathrm{DATE}_i$.

## 6. Result and Discussion

This section presents an analysis of MECAT's data diversity, the analysis of the DATE metric, and a comprehensive evaluation of state-of-the-art models on MECAT.

### 6.1. Data Analysis

**Distribution and Diversity** The t-SNE analysis in Figure 4 highlights MECAT's superior coverage. Regarding *audio* (Figure 4-A/B), MECAT spans the full feature space with distinct internal clusters for pure domains, whereas existing benchmarks remain densely clustered around sound-event regions. Regarding *captions* (Figure 4-C/D), MECAT exhibits significantly richer semantic diversity driven by distinct content categories (speech, music, events) rather than simple length variations.

**Quality Validation** The trustworthiness of MECAT annotations is ensured through a rigorous multi-stage process. The expert models deployed in the pipeline (e.g., CED, Audio Flamingo 2) are architecturally independent with largely uncorrelated error distributions, making concurrent hallucination of the same content statistically unlikely. Combined with LLM-CoT conflict resolution and GLAP filtering, only ~10% of candidate samples survived the full pipeline for release. On this filtered set, two rounds of human evaluation were further conducted. A preference A/B test ($N = 150$) verified that MECAT captions are superior to generic or inaccurate descriptions ($> 94\%$ win rates) and comparable to human-written references (56.9% win rate). Additionally, to directly assess dataset quality, 20 audio-domain profes-

sionals independently rated 700 samples (100 per category) on a 1–5 Relevance (REL) scale to evaluate the relevance between audio and provided captions, where 3 denotes a basic match and 1 denotes completely unrelated. The strict error rate (REL = 1) was bounded at only 3.4% [95% CI: 2.3%–5.1%], and all seven categories scored significantly above 3.0 ($p < 0.001$, Wilcoxon test with Bonferroni correction). Comprehensive details are provided in Appendix F.

### 6.2. Metric Analysis

This section validates our proposed metric, DATE, against the strong baseline FENSE, using an LLM-as-judge method as the upper-bound reference (prompts and reliability analysis in Appendices G and H).

**Qualitative Analysis** The case study in Figure 5-A exposes critical flaws in existing metrics. Lexical-based metrics like BLEU-1 are semantically unreliable, assigning higher scores to Wrong captions than Safe ones. While FENSE improves upon this, it struggles to distinguish high-quality (Right) from vague (Safe) captions, showing a negligible score gap ($\Delta \approx 10$). In contrast, DATE aligns with the clear separation observed in LLM-as-judge scores. DATE demonstrates a clear advantage over FENSE, evidenced by significantly larger median score spans for both Right vs. Wrong (DATE: 46 vs. FENSE: 30) and Right vs. Safe (DATE: 33 vs. FENSE: 24).

**Quantitative Analysis** The Cumulative Distribution Function (CDF) curves in Figure 5-B further quantify discriminative power, where larger inter-curve distances indicate superior performance. Furthermore, a bootstrap stability analysis confirms that DATE produces reliable scores, with Coefficient of Variation (CV) below 1% at full scale for

*Table 2.* Model performance (DATE %) on MECAT-Caption. **Bold** indicates the best performance, and underline indicates the second best. [†] indicates that its previous version (Audio Flamingo 2) was explicitly used in the data construction process.

| Type | Model | Systemic | | Content-Specific | | | | | | Content Unrelated | $Score_{Cap}$ |
| | | | | Speech | | Music | | Sound | | | |
| | | Long | Short | Pure | Mixed | Pure | Mixed | Pure | Mixed | Env | |
|---|---|---|---|---|---|---|---|---|---|---|---|
| Caption -Only | Pengi | 43.5 | 46.8 | 27.2 | 29.5 | 29.3 | 13.1 | 42.8 | 14.6 | 7.1 | 29.4 |
| | EnClap | 48.6 | 53.1 | 30.2 | 31.8 | 17.9 | 15.9 | 48.8 | 15.2 | 6.8 | 31.9 |
| LALM | Phi-4-Multimodal | 42.4 | 44.0 | 26.9 | 31.3 | 14.9 | 24.0 | 28.5 | 18.1 | 13.1 | 30.0 |
| | Kimi-Audio-7B | 49.5 | 54.2 | 30.0 | 31.3 | 27.7 | 16.9 | 43.1 | 16.2 | 7.0 | 32.8 |
| | Baichuan-Audio | 42.6 | 36.5 | 46.0 | 40.4 | 21.3 | 20.7 | 44.8 | 17.7 | 15.1 | 33.7 |
| | Baichuan-Omni | 47.0 | 50.9 | 43.5 | 41.7 | 35.2 | 13.7 | 34.3 | 19.7 | 11.3 | 35.6 |
| | MiMo-Audio | 56.5 | 56.9 | 45.8 | 44.9 | 35.8 | 19.4 | 46.8 | 21.0 | 9.8 | 40.1 |
| | Audio Flamingo 3[†] | 52.5 | 51.5 | 49.3 | 48.8 | 40.4 | 24.8 | 50.6 | 21.9 | 11.5 | 40.4 |
| | Qwen3-Omni | 47.9 | 43.7 | 50.2 | 48.7 | 51.2 | 26.8 | 49.0 | 19.5 | 18.2 | 40.4 |
| | Step-Audio-2-mini | 55.6 | 58.7 | 44.2 | 43.6 | 35.3 | 32.0 | 42.8 | 18.9 | 16.1 | 41.5 |
| | Qwen2.5-Omni 3B | 56.4 | 55.2 | 42.5 | 41.3 | 46.6 | 29.7 | 52.9 | 23.9 | 19.4 | 42.5 |
| | Qwen2.5-Omni 7B | 61.1 | 56.5 | 39.9 | 40.9 | 32.1 | 30.9 | 50.7 | 23.8 | 17.9 | 42.6 |
| | Qwen3-Omni-Flash-1201 | **65.7** | 62.5 | 59.2 | 59.9 | **57.4** | 32.5 | **55.8** | **31.6** | **27.1** | 52.9 |
| | Gemini-2.5-Flash | 65.6 | 63.9 | 57.5 | 57.5 | 52.9 | **41.0** | 52.2 | 28.3 | 22.1 | 51.6 |
| | Gemini-2.5-Pro | 62.3 | 62.4 | 56.6 | 57.5 | 53.6 | 38.7 | 53.4 | 29.9 | 24.0 | 50.6 |
| | Gemini-3-Flash | 63.6 | 61.9 | 59.4 | 60.8 | 43.1 | 32.9 | 51.1 | 29.7 | 25.7 | 51.1 |
| | Gemini-3-Pro | 64.9 | **65.8** | **60.5** | **62.4** | 49.8 | 39.8 | 55.1 | 29.9 | 26.1 | **53.1** |

nearly all evaluation groups, and CV dropping below 3% at 5% sampling. More information could be seen in Appendix F.

**Alignment with Human Judgment** To assess the alignment between DATE and human perception, we utilized the same 150 A/B caption pairs described in Appendix F. Captions preferred by human evaluators received substantially higher DATE scores (Mean: 90.9) compared to non-preferred ones (Mean: 49.3). This significant margin indicates that DATE is strongly correlated with human preferences regarding accuracy and detail, as shown in Table F.7.

### 6.3. Model Performance on MECAT

#### 6.3.1. OVERALL PERFORMANCE

An extensive collection of publicly available models was evaluated, including 17 models for Captioning (15 LALMs, 2 traditional baselines) and 15 LALMs for QA. The evaluated models are strictly categorized into two primary types: traditional Caption-Only models (e.g., Pengi (Deshmukh et al., 2023), EnClap (Kim et al., 2024)) and Large Audio-Language Models (LALMs). The LALMs are further stratified into four architectural subcategories: i) *Audio-focused LALMs* (e.g., Kimi-Audio (Ding et al., 2025), MiMo-Audio (Zhang et al., 2025), Baichuan-Audio (Li et al., 2025), Step-Audio-2-mini (Wu et al., 2025), Audio Flamingo 3 (Ghosh et al., 2026)), ii) *Omni LALMs* (e.g., Qwen2.5-Omni (Xu et al., 2025a), Qwen3-Omni (Xu et al., 2025b), Qwen3-Omni-Flash-1201 (Qwen Team, 2025),

Baichuan-Omni (Li et al., 2024)), iii) *Multimodal LALMs* (Phi-4-Multimodal (Abouelenin et al., 2025)), and iv) the *Gemini series* (Comanici et al., 2025; Google, 2025). Detailed specifications regarding the model architectures and the corresponding prompts utilized in this study are provided in Appendix I.

**Performance on MECAT-Caption.** As shown in Table 2, LALMs demonstrate a substantial advantage over traditional baselines, with scores ranging from 29.4 (Pengi) to 53.1 (Gemini-3-Pro), attributed to superior instruction-following capabilities. Regarding domain stability, a distinct disparity is observed: while performance on Speech tasks remains robust, Music and Sound tasks suffer significant degradation ($10\% \sim 25\%$) when transitioning from Pure to Mixed setting. This suggests a prevalent speech-centric bias in current architectures. Furthermore, the consistently suboptimal performance on Content-Unrelated tasks underscores an urgent need for models to better capture intrinsic sound properties beyond high-level event recognition.

**Performance on MECAT-QA.** The hierarchy in the QA task (Table 3) largely mirrors captioning, yet with a notable shift: the gap between proprietary and open-weight models narrows significantly. Qwen3-Omni (52.3) slightly outperforms both Gemini-2.5-Flash (52.1) and Pro (51.5). At a granular level, we observe a distinct *capability dichotomy*: models excel in Direct Perception and content-based Inference, but degrade significantly on tasks requiring the analysis of intrinsic acoustic properties, such as Quality Assessment and Environment Reasoning. For instance, even

*Table 3.* Model Performance (DATE %) on MECAT-QA. **Bold** indicates the best performance, and underline indicates the second best. [†] indicates that its previous version (Audio Flamingo 2) was explicitly used in the data construction process.

| Model | Perception | Analysis | | Reasoning | | | $\text{Score}_{QA}$ |
|---|---|---|---|---|---|---|---|
| | Direct Perception | Sound Characteristics | Quality Assessment | Environment Reasoning | Inference & Judgment | Application Context | |
| Kimi-Audio-7B | 45.6 | 39.2 | 18.7 | 34.6 | 48.9 | 41.2 | 38.0 |
| Baichuan-Audio | 40.7 | 45.2 | 31.0 | 35.1 | 49.0 | 46.9 | 41.3 |
| Baichuan-Omni | 43.6 | 44.7 | 33.7 | 39.9 | 49.3 | 49.1 | 43.4 |
| Phi-4-Multimodal | 48.4 | 46.3 | 34.7 | 40.2 | 49.3 | 48.7 | 44.6 |
| MiMo-Audio | 59.3 | 49.3 | 24.9 | 39.1 | 52.7 | 46.2 | 45.2 |
| Step-Audio-2-mini | 57.7 | 54.3 | 37.2 | 39.2 | 48.9 | 48.0 | 47.6 |
| Audio Flamingo 3[†] | 53.8 | 50.2 | 36.0 | 43.0 | 54.5 | 49.6 | 47.8 |
| Qwen2.5-Omni 3B | 55.7 | 53.2 | 38.6 | 41.1 | 51.8 | 50.8 | 48.5 |
| Qwen2.5-Omni 7B | 57.8 | 52.9 | 39.1 | 44.0 | 53.2 | 50.8 | 49.6 |
| Qwen3-Omni | **61.7** | 54.6 | **39.3** | 45.0 | 56.9 | 56.1 | **52.3** |
| Qwen3-Omni-Flash-1201 | 48.0 | 45.9 | 29.5 | 45.6 | 56.7 | 54.8 | 46.7 |
| Gemini-2.5-Flash | 56.3 | **55.3** | 37.7 | 46.8 | **58.6** | **58.0** | 52.1 |
| Gemini-2.5-Pro | 55.5 | 54.4 | 37.7 | **47.6** | 57.3 | 56.6 | 51.5 |
| Gemini-3-Flash | 54.3 | 51.1 | 34.1 | 47.2 | 57.2 | 57.0 | 51.0 |
| Gemini-3-Pro | 55.5 | 45.5 | 25.8 | 44.0 | 53.2 | 52.0 | 46.0 |

top models fail to exceed a score of 40 in Quality Assessment, confirming that current LALMs prioritize high-level semantics over nuanced acoustic interpretation.

### 6.3.2. IN-DEPTH ANALYSIS: LALM BOTTLENECKS

While standard metrics rank models, they often mask underlying behavioral nuances. This study utilizes the Captioning task to probe two critical bottlenecks: discriminability limitations and robustness against hallucination.

**The Critical Role of Discriminability.** A granular analysis of the Pure Speech subset using DATE (Appendix J) underscores the interplay between Semantic Similarity and Discriminability (note that for other subtasks, discriminability could be derived from the Similarity and DATE scores in Table 2 and Appendix K). Top-tier proprietary models—notably Qwen3-Omni-Flash-1201 and the Gemini series—dominate both metrics, pairing superior similarity with exceptional discriminability (above 77.0). Notably, the open-source Qwen3-Omni demonstrates that robust discriminative power (64.7) can compensate for moderate similarity to secure a top-tier ranking. This suggests that while elite models excel globally, strong acoustic discriminability is a primary driver for high-quality captioning.

**Hallucination in Silent Segments.** Qualitative evaluation of silent segments (Appendix L) reveals a significant robustness issue. While models like Qwen3-Omni-Flash-1201 and Gemini-2.5 correctly identify silence, many LALMs over-generate, resulting in hallucinations of specific but unrelated text (e.g., "*I'm gonna be a daddy*" or "*Thank you*"). This tendency to over-generate exposes a fundamental vul-

nerability, underscoring the necessity for improved rejection mechanisms in future architectures.

### 6.4. Limitations

While MECAT advances fine-grained audio evaluation, several limitations should be acknowledged.

**Audio scope.** The 10-second clip duration prioritizes dense, local-level supervision over long-range temporal reasoning; MECAT is thus complementary to long-audio benchmarks such as LongAudioBench rather than a substitute.

**Annotation quality.** Although the multi-expert pipeline and rigorous filtering substantially mitigate upstream model errors, residual noise from expert hallucinations cannot be entirely eliminated, particularly in non-speech categories where annotation difficulty is higher.

**DATE metric.** DATE inherits fundamental limitations of embedding-based similarity, including potential insensitivity to entity swaps, and its cross-sample discriminability component is most informative when the test set contains diverse audio clips; on highly homogeneous corpora, its discriminative power may diminish.

## 7. Conclusion and Future Work

In this work, we introduced MECAT, a Multi-Experts Constructed Benchmark leveraged by Chain-of-Thought reasoning to advance fine-grained audio understanding in Captioning and QA tasks. Complementing this, we proposed DATE, a novel metric tailored to penalize vague terminology and incentivize detailed, discriminative descriptions.

## Acknowledgements

We specially thank Jiahao Mei (Shanghai Jiao Tong University), who contributed to the subjective evaluation during the revision process. His work on experimental design and data collection greatly improved the quality of this paper. We also thank all annotators who participated in the evaluation, and the anonymous reviewers for their constructive feedback.

## Impact Statement

This work contributes a benchmark and metric for audio understanding evaluation. We discuss the potential societal implications below.

**Positive impact.** MECAT and DATE are designed to advance the evaluation of audio understanding systems, which can yield positive societal benefits such as improved accessibility for hearing-impaired individuals and enhanced context-aware environmental awareness.

**Potential risks.** Fine-grained audio captioning systems could potentially be applied to automatically describe audio recordings containing private or sensitive information, raising concerns about privacy leakage and surveillance misuse.

**Mitigation.** MECAT itself is constructed exclusively from publicly available, Creative Commons-licensed data (ACAV100M), ensuring the benchmark does not contain private content. For downstream deployment of fine-grained audio captioning systems, we recommend that practitioners implement privacy-preserving safeguards, including sensitive content detection and filtering before caption generation, anonymization of personally identifiable information in outputs, and adherence to informed consent principles when processing non-public audio data.

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

## A. LLM usage

In the preparation of this manuscript, we utilized Gemini 2.5 Pro model (accessed in July 2025) primarily for proofreading and grammatical corrections. The tool was used to improve the clarity and readability of the text. All authors have reviewed and edited the final manuscript and take full responsibility for its content.

## B. Vocabulary Size of Audio Captioning TestSet

This section introduces vocabulary size comparison to demonstrate the lexical diversity of MECAT-Caption. The following table indicates that the vocabulary size of MECAT-Caption contains about 4-17 times more words than the existing dataset.

*Table B.1.* Vocabulary Size of Audio Captioning Test Set

| Dataset | # Vocab |
|---|---|
| AudioCaps | 5,581 |
| AudioCaps-Enhanced | 1,260 |
| AutoACD | 3,517 |
| Clotho | 1,852 |
| StrongDescriber | 2,726 |
| LPMusicCaps-MTT | 1,666 |
| **MECAT-Caption** | **22,595** |

# C. Deployed Acoustic Models in Processing Pipeline

This section introduces the acoustic models deployed in our processing pipeline. These models are categorized into Content-Specific models (including Speech, Music, and Sound analysis) and Content-Unrelated models (Environment analysis), each designed to handle different aspects of audio understanding tasks.

*Table C.1.* Deployed Acoustic Models in Processing Pipeline

| Category | Subcategory | Model | Analsysis Task |
|---|---|---|---|
| Content Specific | Speech | Speechbrain-ECAPA(Ravanelli et al., 2024)
Whisper Large v2 (Radford et al., 2023)
Pyannote-SD 3.1 (Bredin, 2023)
Emotion2Vec (Ma et al., 2024)
Audeering-DSER (Wagner et al., 2023)
Audeering-AGR (Burkhardt et al., 2023)
CommonAccent (Zuluaga-Gomez et al., 2023) | Language Recognition
Auto Speech Recognition
Speaker Diariazation
Speaker Emotion Recognition
Dimentional Speaker Emotion Recogintion
Age and Gender Recoginition
English Accent Recognition |
| | Music | Music Structure Analyzer (Kim & Nam, 2023)
Music2Emo (Kang & Herremans, 2025)
MERT (Yizhi et al., 2023)
ByteSep (Kong et al., 2021)
Audio Flamingo 2 (Ghosh et al., 2025) | Tempo & Structure
Emotion (Sentiment Polarity and Mood)
Musical Descriptor
Music Seperation
AudioLLM |
| | Sound | CED (Dinkel et al., 2024a) | Sound Event Recognition |
| Content Unrelated | Environment | DNSMOS (Reddy, 2021; 2022)
NISQA V2.0 (Mittag et al., 2021)
SHAART (Hawley, 2023) | Noisy Speech Quality Assessment
Multidimensional Speech Quality Assessment
Reverberation |

# D. LLM Audio Analysis Synthesis Prompts

Act as an expert audio analysis synthesizer to process multi-model JSON outputs through this workflow

Step 1: Multi-Domain Data Specifications

1.1 Multi-Domain Input Integration

    a) Speech: Speech recognition, speech emotion, speaker diarization and so on

    b) Music: Structure analysis, technical descriptors, emotion and so on

    c) Sound: Event detection timestamps, classifications

    d) Environment: Acoustic characteristics, interference markers

    e) Meta-info: Title and description of original video where audio clip was extracted

1.2 Data Integrity Challenges

    a) Missing fields

    b) Contradictory model outputs

    c) Confidence score variances

Step 2: Technical and Analytical Limitations

2.1 Model and System Constraints

    a) No speech recognition ability in audio captioning models (e.g., audio-flamingo variants)

    b) Accuracy disparities across the analyzed domains

    c) Potential conflicting information between models

2.2 Audio Content Heterogeneity

    a) Hybrid audio types (e.g., speech, music, sound-event, environment)

    b) Variable audio properties (e.g., clip lengths or quality)

    c) Reliable topic or domain, but absent or non-relevant details in Meta-info

Step 3: Audio Analysis Workflow

3.1 Salient Feature Identification

  3.1.1 Identify dominant characteristics of this audio:

What makes this specific audio clip unique according to the analysis? Examples include:

    a) Specific spoken phrases

    b) Dominant musical styles or moods

    c) Significant sound events

    d) The overall acoustic scene

    e) Notable quality issues

    f) Complex interplay of elements

3.1.2 Supporting Evidence Extraction:

Gather the key details describing these salient features from the relevant JSON fields

3.2 Synthesis Rules

3.2.1 Generation Rules:

a) Critically weigh evidence from different fields, considering inaccuracies or conflicts and accounting for domain-specific limitations

b) Prioritize information most reliable or central to the audio's character based on overall data patterns

c) Carefully identify conflicting information between fields and avoid mentioning conflicting aspects in the final caption. Focus only on consistent and unopposed information. Do not invent details not present in the data

d) Crucial Constraint:

- The final generated text must strictly describe only the analyzed content of the audio segment itself
- It must not refer to the topic, title, description, or inferred subject matter from the overall video metadata
- Avoid phrases like "in a clip from a video about..." or similar references to the source video's topic
- Prohibit using parentheses to provide detailed explanation in any output, e.g., Moderate tempo (88 BPM)

3.2.2 Perspective Rules:

ALL answers must be created from the perspective of someone who ONLY LISTENED to the

audio without any technical/model references or quantitative metrics (e.g., BPM, MOS, etc.)

3.2.3 Evaluation Rules:

Assign a confidence level (High or Low) based on the following aspects:

a) Consistency: Are the different analyses in the JSON generally consistent or contradictory? High consistency increases confidence

b) Completeness: Is key information present? (Fewer gaps = higher confidence)

c) Clarity: How clearly does the consistent data point to the audio's nature? (Less ambiguity in reliable data = higher confidence)

d) Metadata Context Usefulness: How relevant and useful was the overall video metadata in confirming or contextualizing findings from the clip's direct analysis?

3.3 Caption Development Framework

3.3.1 Systematic Caption

a) Short ($<$ 15 words):
- Protocol: Primary domain characteristics + Most prevalent characteristic from cross-model correlation
- Example: Blues guitar performance at live concert with audience reactions

b) Long (1-2 sentences):
- Protocol: Primary domain + significant secondary elements + notable quality factors
- Example: A live concert recording featuring guitar with crowd cheers, despite occasional microphone static

3.3.2 Content-Focused Caption

a) Speech:
- Protocol: ASR content + paralinguistic context

- Example: Two speakers discussing jazz history, with piano accompaniment

b) Music:
  - Protocol: Technical descriptors + performance context
  - Example: Upbeat electronic track with distant traffic noise

c) Sound:
  - Protocol: Event taxonomy + spatial relationships
  - Example: Office environment with printer hum and keyboard typing, mild echo present

### 3.3.3 Content-Unrelated Caption

a) Environment:
  - Protocol: Acoustic properties + interference profile
  - Example: Studio recording with noticeable background interference

### 3.3.4 Caption Variants

a) Lexical substitution (WordNet-based synonyms)

b) Structural reordering (active/passive voice)

c) Descriptive equivalence ('crowd cheers' → 'audience applause')

### 3.3.5 Null Handling

When no domain-specific elements are detected:

a) Use explicit 'None' declaration in content field

b) Generate null statement variants (e.g., 'No discernible speech content', 'Musical elements appear absent')

## 3.4 Question-Answering Design

### 3.4.1 Content Categories

Include questions across:

a) Direct Perception (sound type, volume, duration)

b) Sound Characteristics (timbre, rhythm, frequency characteristics)

c) Environmental Perception (recording setting, echo, background noise)

d) Quality Assessment (clarity, interference factors)

e) Inference and Judgment (sound source, generation method, object properties)

f) Application Context (use cases, semantic meaning)

### 3.4.2 Difficulty Levels

Include a mix of:

a) Basic: Direct descriptive questions (e.g., 'What sound is heard?')

b) Intermediate: Analytical questions (e.g., 'What are the characteristics of this sound?')

c) Advanced: Inferential questions (e.g., 'In what environment was this recorded?')

d) Complex: Comprehensive judgment questions (e.g., 'Based on the sound, what is the most likely material?')

3.4.3 Question Distribution

Basic (25%) — Intermediate (35%) — Advanced (25%) — Complex (15%)

3.4.4 Question Variety

Include:

a) Ensure questions cover all listed categories

b) Avoid repetitive question patterns or formats

c) Include both yes/no questions and open-ended questions

d) Include some questions about what is NOT present in the audio

e) Include some comparative questions (e.g., 'Does this sound more like X or Y?')

3.4.5 Cognitive Levels

Include:

a) Include questions requiring simple recognition

b) Include questions requiring analysis of components

c) Include questions requiring synthesis of information

d) Include questions requiring evaluation or judgment

Step 4: Structured Output Specification (JSON Format)

Confidence: High/Low

Possible Conflicts: None or list of conflicting fields

Reasoning: 2-3 line evaluation considering model consensus and data quality

Short-Caption: Single-sentence essence

Short-Caption-Variants-1: Paraphrased version 1

Short-Caption-Variants-2: Paraphrased version 2

Main-Caption: Integrated summary

Main-Caption-Variants-1: Paraphrased version 1

Main-Caption-Variants-2: Paraphrased version 2

Speech-Captions: Speech-focused analysis or NONE

Speech-Caption-Variants-1: Paraphrased version 1

Speech-Caption-Variants-2: Paraphrased version 2

Music-Captions: Music-focused analysis or NONE

Music-Caption-Variants-1: Paraphrased version 1

Music-Caption-Variants-2: Paraphrased version 2

Sound-Captions: Sound-focused analysis or NONE

Sound-Caption-Variants-1: Paraphrased version 1

Sound-Caption-Variants-2: Paraphrased version 2

Environment-Caption: Environment-focused analysis

Environment-Caption-Variants-1: Paraphrased version 1

Environment-Caption-Variants-2: Paraphrased version 2

QA-Pair-1-id: 1 or None

QA-Pair-1-difficulty: basic, intermediate, advanced, or complex

QA-Pair-1-category: direct perception,sound characteristics, environmental perception,quality assessment, inference judgment,or application context

QA-Pair-1-question: question content

QA-Pair-1-answer: answer content

QA-Pair-2-id: 2 or None

QA-Pair-2-difficulty: basic, intermediate, advanced, or complex

QA-Pair-2-category: direct perception,sound characteristics, environmental perception,quality assessment, inference judgment,or application context

QA-Pair-2-question: question content

QA-Pair-2-answer: answer content

// ... 3 more QA pairs following the same pattern

# E. Embedding-level TF-IDF Calculation for DATE

## E.1. Rationale for Embedding-level TF-IDF

Classific term frequency-inverse document frequency (TF-IDF) relies on discrete, hard-count token occurrences to calculate term frequency (TF). However, in natural language, the semantic relationship between words (e.g., "dog" and "canine") is lost when treating them as independent tokens.

The Discriminability based Audio Task Evaluation (DATE) metric enhances the representational power of sentence embeddings by incorporating an Embedding-level TF-IDF weighting scheme. This method leverages the semantic information encoded in word embeddings to compute a soft, non-integer Term Frequency, thereby improving the quality of the resulting sentence representation for downstream similarity and discrimination calculations.

## E.2. Method of Calculation

The Embedding-level TF-IDF weight for a word $w$ in a sentence $s$ is calculated as the product of its semantic-aware Term Frequency ($\text{TF}_{\text{emb}}$) and its Inverse Document Frequency ($\text{IDF}_{\text{emb}}$):

$$\text{TF-IDF}_{\text{emb}}(w, s) = \text{TF}_{\text{emb}}(w, s) \times \text{IDF}_{\text{emb}}(w) \tag{E.1}$$

The calculation proceeds in three main steps:

**Semantic-aware Term Frequency ($\text{TF}_{\text{emb}}$)** Instead of counting a word's exact occurrences (which would result in an integer count), $\text{TF}_{\text{emb}}$ is calculated by measuring the average similarity of the word's embedding to the embeddings of all other words within the same sentence. This accounts for semantic context and relatedness.

- **Word Embeddings:** Word-level embeddings are generated for all tokens in the corpus with the Sentense-Bert.

- **Word-to-Word Similarity Matrix (WordSim):** A full WordSim matrix is computed by taking the dot product of the $L_2$-normalized embeddings ($E$) of all unique non-padding words in the batch.
$$\text{WordSim} = \text{Normalize}(E) \cdot \text{Normalize}(E)^T \tag{E.2}$$

- **Term Frequency Calculation:** The semantic Term Frequency for a word $w_i$ in a sentence $s$ is calculated by summing the squared similarity scores with all other words $w_j$ in that sentence (excluding special tokens like `[CLS]` and `[SEP]`):
$$\text{TF}_{\text{emb}}(w_i, s) = \sum_{w_j \in s, j \neq i} \text{WordSim}(i, j)^2 \tag{E.3}$$

- **Result:** This calculation yields a non-integer $\text{TF}_{\text{emb}}$ value, where words semantically central to the sentence receive a higher score.

**Embedding-aware Inverse Document Frequency ($\text{IDF}_{\text{emb}}$)** The Inverse Document Frequency (IDF) component measures a word's uniqueness across the entire document corpus. In the Embedding-level approach, the document frequency (DF) is calculated based on the similarity of a unique word's embedding to the embeddings of all tokens across all documents.

- **Word-to-Document Similarity Matrix (Word2DocSim):** A Word2DocSim matrix is calculated by taking the dot product between the unique word embeddings and the embeddings of all tokens (word pieces) in the corpus.

- **Document Frequency Accumulation:** The document frequency (DF) for a word $w$ is accumulated across the entire corpus by summing the squared Word2DocSim values. This accumulation is performed sentence-by-sentence based on token availability.

- **IDF Calculation:** The final $\text{IDF}_{\text{emb}}$ is calculated using a log-normalization formula, where $N$ is the corpus size:
$$\text{IDF}_{\text{emb}}(w) = \log\left(\frac{N+1}{\text{DF}(w)+1}\right) + 1 \tag{E.4}$$

**Final Weighting and Normalization** The $\text{TF}_{\text{emb}}$ is multiplied by the $\text{IDF}_{\text{emb}}$ to get the final raw TF-IDF weight for each non-special token. These weights are then normalized to ensure stability.

# F. Supplementary Validation Studies

This section provides comprehensive validation of both the MECAT dataset and its evaluation metrics. We first validate data quality through automated pipeline filtering and two rounds of human evaluation (Section F.1), and then verify metric reliability from three complementary perspectives: scoring bias, stability, and alignment with human judgments (Section F.2).

## F.1. Data Quality Validation

The trustworthiness of MECAT annotations relies on a rigorous multi-stage pipeline. The expert models deployed (e.g., CED, Audio Flamingo 2) are architecturally independent with largely uncorrelated error distributions, making concurrent hallucination of the same content statistically unlikely. Through LLM-CoT domain conflict resolution and GLAP filtering, only ~10% of candidate samples from the CC-licensed source data survived the full pipeline for release. On this filtered set, two rounds of human evaluation were conducted to validate data quality from complementary perspectives: annotation scheme consistency with human judgment and annotation result quality at scale.

### F.1.1. HUMAN PREFERENCE A/B TEST

A Human Preference A/B test ($N = 150$ caption pairs spanning all domains) was conducted to verify that the annotation scheme produces outputs consistent with human judgment. Evaluators were instructed to select the better caption based on the stringent criterion: "accuracy first, then level of detail."

MECAT references (A) were compared against three opponent types (B) to probe specific quality aspects: Safe Captions (generic descriptions, testing discriminability); Wrong Captions (factually incorrect references, testing accuracy); and Human References (expert-written ground truth, establishing the quality ceiling).

The results are presented in Table F.1. MECAT references were strongly favored over both Safe and Wrong captions ($> 94\%$ win rates), confirming the pipeline's effectiveness in mitigating vague or factually incorrect outputs. Crucially, the quality of MECAT references was found to be statistically on par with human-written references (56.9% win rate, as the 95% confidence interval spans 50%).

*Table F.1.* Human Preference A/B Validation Results ($N = 150$ pairs). Win rates indicate preference for the MECAT Reference (A).

| Opponent Type (B) | Size | Reference (A) Win Rate | 95% CI (Wilson) |
|---|---|---|---|
| Overall | 150 | 82.7% | [75.8%, 87.9%] |
| Safe Captions | 52 | 94.2% | [84.4%, 98.0%] |
| Wrong Captions | 47 | 97.9% | [88.9%, 99.6%] |
| Human References | 51 | **56.9%** | [43.3%, 69.5%] |

### F.1.2. EXPERT SUBJECTIVE EVALUATION

To directly assess the quality of annotation results at scale, we recruited 20 audio-domain professionals to conduct an independent human audit. For each audio clip, the benchmark provides captions from multiple perspectives, including a global description and domain-specific descriptions from the speech, music, sound, and environment dimensions. Each participant listened to the original audio and judged whether these captions collectively captured the key details of the audio content, assigning a 1–5 Relevance (REL) score as defined in Table F.2.

*Table F.2.* REL scoring scale definition for expert subjective evaluation.

| Score | Interpretation |
|---|---|
| 5 | Perfect match – captions precisely describe all audio details |
| 4 | High match – captions capture nearly all details; only minor omissions |
| 3 | Basic match – captions reflect the general intent of the audio |
| 2 | Low match – key details missing or incorrect |
| 1 | No match – captions are completely unrelated to the audio |

**Overall Results.** Across 700 samples (100 per category), 85.3% scored $\geq 3$ (acceptable or better), 67.3% achieved high

accuracy (scores 4–5), and the mean REL was 3.81. The strict label error rate (REL = 1) was bounded at only 3.4% [95% CI: 2.3%–5.1%]. Scores were significantly above the basic-match baseline of 3.0 (Wilcoxon $p < 0.001$; rank-biserial $r = 0.714$; Cohen's $d = 0.734$).

**Per-Category Results.** Per-category results are presented in Table F.3. All seven categories passed the Wilcoxon signed-rank test ($p < 0.001$) against the 3.0 baseline after Bonferroni correction ($\alpha = 0.05/7$).

*Table F.3.* Per-category human audit results (H$_1$: REL > 3.0, 100 samples each).

| Category | Description | Mean REL | Acceptable ($\geq 3$) | Cohen's $d$ | Rank-biserial $r$ |
|----------|-------------|----------|----------------------|-------------|-------------------|
| 00A | Sound Effects Only | 3.45 | 74.0% | 0.365 | 0.422 |
| 0M0 | Music Only | 3.77 | 80.0% | 0.644 | 0.648 |
| 0MA | Music + Sound Effects | 3.57 | 79.0% | 0.453 | 0.511 |
| S00 | Speech Only | 4.10 | 95.0% | 1.247 | 0.935 |
| S0A | Speech + Sound Effects | 3.89 | 88.0% | 0.838 | 0.788 |
| SM0 | Speech + Music | 4.07 | 94.0% | 1.250 | 0.915 |
| SMA | Speech + Music + Sound Effects | 3.85 | 87.0% | 0.790 | 0.744 |

**Domain Variance.** We observed meaningful scoring variance across audio domains: Speech (mean 3.98) > Music (3.81) > Sound (3.69). The difference between Speech and Sound domains is statistically significant (Mann–Whitney $U$, $p = 1.37 \times 10^{-3}$), with a small but measurable effect size (Cliff's $\delta = 0.125$; Cohen's $d = 0.267$). This quantifies the added difficulty of non-speech event alignment, particularly in categories where coarse metadata may not fully capture fine-grained acoustic details.

*Table F.4.* Model performance (Score$_{\text{Cap}}$) across different weighting scenarios. Kendall's $\tau = 0.92$ between Original and the other two settings.

| Model | Original (6:3:1) | | Audio-Centric (2:4:4) | | Equal (1:1:1) | |
|-------|------|------|------|------|------|------|
| | Score | Rank | Score | Rank | Score | Rank |
| Gemini-3-Pro | 53.10 | 1 | 50.14 | 2 | 51.09 | 2 |
| Qwen3-Omni-Flash-1201 | 52.90 | 2 | 50.39 | 1 | 51.20 | 1 |
| Gemini-2.5-Flash | 51.60 | 3 | 49.08 | 3 | 49.82 | 3 |
| Gemini-3-Flash | 51.10 | 4 | 47.80 | 5 | 48.91 | 5 |
| Gemini-2.5-Pro | 50.60 | 5 | 48.34 | 4 | 49.04 | 4 |
| Qwen2.5-Omni 7B | 42.60 | 6 | 41.88 | 7 | 42.21 | 6 |
| Qwen2.5-Omni 3B | 42.50 | 7 | 41.94 | 6 | 42.14 | 7 |
| Step-audio-2-mini | 41.50 | 8 | 39.54 | 8 | 40.16 | 8 |
| Qwen3-Omni | 40.40 | 9 | 38.14 | 10 | 38.82 | 11 |
| Audio Flamingo 3 | 40.40 | 10 | 38.16 | 9 | 38.94 | 9 |
| MiMo-Audio-Instruct | 40.10 | 11 | 38.06 | 11 | 38.84 | 10 |
| Baichuan-Omni | 35.60 | 12 | 33.01 | 12 | 33.91 | 12 |
| Baichuan-Audio-Instruct | 33.70 | 13 | 31.39 | 14 | 32.30 | 14 |
| Kimi-Audio-7B-Instruct | 32.80 | 14 | 32.34 | 13 | 32.59 | 13 |
| Phi-4-Multimodal-Instruct | 30.00 | 15 | 28.88 | 15 | 29.29 | 15 |

### F.2. Metric Validation

This subsection validates the reliability of the proposed evaluation metrics from three complementary perspectives: whether the scoring weights introduce systematic bias, whether the DATE metric produces stable scores across varying test conditions, and whether DATE aligns with human judgments.

### F.2.1. SCORECAP WEIGHTING SENSITIVITY ANALYSIS

To verify that the heuristic weighting scheme for $Score_{Cap}$ does not bias model rankings toward speech-dominant models, we conducted a sensitivity analysis across 15 LALMs under three weighting scenarios: the Original data-driven weights (Speech/Music/Sound = 6:3:1), an Audio-Centric setting (2:4:4) that emphasizes non-speech domains, and an Equal setting (1:1:1). As shown in Table F.4, model rankings remain highly stable across all three scenarios, with Kendall's $\tau = 0.92$ between the Original and the other two settings. This confirms that top-performing models excel through comprehensive acoustic understanding rather than exploiting the speech-weighted distribution.

### F.2.2. DATE BOOTSTRAP STABILITY ANALYSIS

To assess the stability of DATE scores under varying test set sizes, we conducted a bootstrap stability analysis using Gemini-3.0-Pro outputs. Speech, Music, and Sound are each evaluated separately on Pure (single audio type) and Mixed (overlapping types) subsets with independent reference pools, yielding nine evaluation groups in total (including Long, Short, and Environment). For each group at each sampling ratio, 50 bootstrap iterations were performed, each fully re-computing DATE including the discrimination-rank component.

**Full-Scale Stability.** Table F.5 summarizes bootstrap results at 100% sampling. Eight out of nine groups exhibit a Coefficient of Variation (CV) below 1%, with only Pure-Sound (CV = 1.04%, $N = 848$) slightly exceeding this threshold due to its smaller pool size. The maximum deviation across all groups is bounded at $\pm 0.014$.

*Table F.5.* Full-data (100%) bootstrap summary across nine evaluation groups (50 iterations each).

| Group | Size | DATE Mean | CV (%) | 95% CI | Max Dev |
|---|---|---|---|---|---|
| Long | 20,052 | 0.6486 | 0.14 | [0.647, 0.650] | ±0.002 |
| Short | 20,052 | 0.6581 | 0.16 | [0.656, 0.660] | ±0.003 |
| Pure-Speech | 7,839 | 0.6049 | 0.29 | [0.602, 0.607] | ±0.004 |
| Mixed-Speech | 8,593 | 0.6236 | 0.31 | [0.620, 0.627] | ±0.004 |
| Pure-Music | 2,593 | 0.4967 | 0.77 | [0.490, 0.503] | ±0.008 |
| Mixed-Music | 8,593 | 0.3988 | 0.46 | [0.395, 0.402] | ±0.004 |
| Pure-Sound | 848 | 0.5520 | 1.04 | [0.542, 0.565] | ±0.014 |
| Mixed-Sound | 8,593 | 0.2992 | 0.94 | [0.294, 0.304] | ±0.007 |
| Environment | 20,052 | 0.2613 | 0.58 | [0.259, 0.264] | ±0.003 |

**Convergence Across Subset Sizes.** Table F.6 reports the CV across different sampling ratios. CV drops below 3% at 5% sampling for seven out of nine groups. Even for the smallest group (Pure-Sound, $N = 848$), CV reaches below 6% at 5% sampling. At $\geq 10\%$ sampling, all groups except Pure-Sound and Pure-Music fall below 2.5%.

*Table F.6.* CV (%) across subset sizes. CV@$x$% denotes CV from 50 bootstrap iterations each drawing $x$% of full data with replacement.

| Group | CV@1% | CV@5% | CV@10% | CV@20% | CV@50% | CV@100% |
|---|---|---|---|---|---|---|
| Long | 1.79 | 0.61 | 0.54 | 0.32 | 0.22 | 0.14 |
| Short | 1.60 | 0.58 | 0.55 | 0.36 | 0.22 | 0.16 |
| Pure-Speech | 3.14 | 1.37 | 0.92 | 0.68 | 0.40 | 0.29 |
| Mixed-Speech | 3.53 | 1.22 | 0.88 | 0.67 | 0.47 | 0.31 |
| Pure-Music | 10.65 | 4.10 | 3.22 | 2.29 | 1.32 | 0.77 |
| Mixed-Music | 5.86 | 2.69 | 2.28 | 1.37 | 0.67 | 0.46 |
| Pure-Sound | 18.62 | 5.85 | 4.61 | 3.22 | 1.68 | 1.04 |
| Mixed-Sound | 9.60 | 4.36 | 3.11 | 2.77 | 1.28 | 0.94 |
| Environment | 5.60 | 2.92 | 1.90 | 1.32 | 0.98 | 0.58 |

These results confirm that DATE produces stable, scale-invariant scores across all evaluation groups, supporting its reliability as an evaluation metric for fine-grained audio captioning.

F.2.3. ALIGNMENT OF DATE WITH HUMAN JUDGMENTS

The alignment of the DATE metric with human preferences was assessed using the same 150 A/B caption pairs from Section F.1.1, where listener choices established the human "gold standard".

For this validation, the human-preferred caption and the non-preferred caption from each A/B pair were separately treated as a candidate hypothesis. The DATE score was then computed for each candidate against the audio segment's original ground truth reference.

A comparison of mean DATE scores, presented in Table F.7, reveals a substantial difference: captions selected by human evaluators received substantially higher mean DATE scores (90.9) compared to non-preferred captions (49.3). This large margin ($\Delta \approx 41.6$) robustly demonstrates that DATE is highly correlated with human preference regarding accuracy and detail, thereby validating its utility as a fine-grained evaluation tool.

*Table F.7.* Alignment between DATE Score and Human Preference ($N = 150$ A/B pairs).

| Caption Group | DATE |
|---|---|
| Human-Preferred Captions | **90.9** |
| Non-Preferred Captions | 49.3 |

# G. LLM-as-Judge Prompts in evaluation

This section provides the prompt template required for LLM-as-Judge method. The evaluation *tasks* primarily include audio captioning and audio question-answering. In the template, the *description*, *subtask*, and *scoring_aspects* parameters can be referenced from the corresponding columns in the task table above, while *ref_texts* represents the samples to be evaluated.

**G.1.** Evaluation Prompt Template

You are tasked with evaluating if a set of candidate {tasks} responses accurately addresses the same audio as a reference set of answers. You will focus on the {description} for the subtask '{subtask}'.

**Evaluation Steps:**

a) First, carefully compare the candidate answers with the reference answers

b) Assess the accuracy and precision of how the audio characteristics are captured in the responses, then provide a 0-10 fine-grained score:
   - 10 = perfect match with the reference content
   - 0 = completely wrong

c) Provide detailed scoring reasoning, explaining why you gave this score

**Scoring Aspects:** {scoring_aspects}

**Score rubric (0-10 Scale):**

- points 9-10: Excellent - Highly accurate, comprehensive, well-expressed

- points 8:    Very Good - Accurate with minor gaps, clear expression

- points 7:    Good - Mostly accurate, some missing details

- points 6:    Acceptable - Basic accuracy, meets minimum HIGH standard

- points 4-5:  Below Standard - Some correct elements but major issues

- points 2-3:  Poor - Limited accuracy, significant problems

- points 0-1:  Very Poor - Major errors or completely incorrect

**You need to evaluate the following sample:** {ref_texts}

**Please return JSON-formatted evaluation results for the sample.**

Return format (strict JSON array):

sample_id: sample ID

subtask: subtask_name

fine_score: <numerical value 0-10>

reasoning: detailed scoring rationale, including comparative analysis with reference answers

*Table G.1.* Category, subcategory, descriptions and scoring aspects of captioning evaluation with LLM-as-Judge method

| Category | Subcategory | Description & Scoring Aspects |
|---|---|---|
| Systemic | Short | **Description:** quality of short audio descriptions
**Scoring Aspects:**
 a) accuracy of core content capture (most important)
 b) conciseness and completeness of expression
 c) semantic consistency with reference descriptions |
| | Long | **Description:** quality of detailed audio descriptions
**Scoring Aspects:**
 a) comprehensiveness and richness of description details
 b) accuracy of detailed descriptions
 c) logical structure and expression coherence |
| Content-Specific | Speech | **Description:** accuracy of speech content recognition
**Scoring Aspects:**
 a) accuracy rate of speech content recognition
 b) accurate description of speaker characteristics (gender, accent, etc.)
 c) description of speech quality and environment |
| | Music | **Description:** quality of music content description
**Scoring Aspects:**
 a) accuracy of music type, style, and rhythm identification
 b) identification of instruments and musical elements
 c) description of musical emotion and atmosphere |
| | Sound | **Description:** accuracy of sound event identification
**Scoring Aspects:**
 a) accurate identification and classification of sound sources
 b) description of sound occurrence timing and duration
 c) description of sound intensity, pitch and other characteristics |
| Content-Unrelated | Environment | **Description:** accuracy of environment and recording quality description
**Scoring Aspects:**
 a) identification of recording environment (indoor/outdoor, space size, etc.)
 b) assessment of audio technical quality (distortion, noise, etc.)
 c) description of environmental atmosphere and background characteristics |

*Table G.2.* Category, subcategory, descriptions and scoring aspects of question-answering evaluation with LLM-as-Judge method

| Category | Subcategory | Description & Scoring Aspects |
|---|---|---|
| Perception | Direct Perception | **Description:** accuracy of direct audio content identification
**Scoring Aspects:**
  a) correct identification of primary audio elements (most important)
  b) accurate detection of presence/absence of specific sounds
  c) precise recognition of obvious audio features and events |
| Analysis | Sound Characteristics | **Description:** quality of sound property analysis
**Scoring Aspects:**
  a) accurate description of sound attributes (pitch, volume, timbre, etc.)
  b) correct identification of sound sources and their properties
  c) precise characterization of audio dynamics and patterns |
|  | Quality Assessment | **Description:** accuracy of audio quality evaluation
**Scoring Aspects:**
  a) correct assessment of technical audio quality (clarity, distortion, etc.)
  b) accurate evaluation of recording conditions and fidelity
  c) appropriate judgment of audio production quality |
| Reasoning | Environment Reasoning | **Description:** quality of environmental context inference
**Scoring Aspects:**
  a) accurate inference of recording location and setting
  b) correct identification of spatial and acoustic properties
  c) logical deduction of environmental factors affecting audio |
|  | Inference Judgment | **Description:** accuracy of complex audio analysis and reasoning
**Scoring Aspects:**
  a) correct interpretation of implicit audio information
  b) accurate temporal reasoning and sequence understanding
  c) logical inference of causality and relationships between audio elements |
|  | Application Context | **Description:** relevance and appropriateness of contextual understanding
**Scoring Aspects:**
  a) accurate understanding of audio's intended purpose or context
  b) appropriate application of domain-specific knowledge
  c) correct interpretation of cultural, social, or professional context |

## H. Validation of LLM-as-Judge as a reference metric

To validate the effectiveness of LLM-as-Judge as a reference metric, we assessed its performance on three distinct sets of responses with varying quality levels: Right (detailed and accurate rephrasings of the ground-truth reference), Safe (generic, vague descriptions, e.g., "A man is speaking" for all speech-only audio), and Wrong (factually incorrect references randomly selected from other samples). As shown in following table, our analysis confirms that LLM-as-Judge method serves as a reliable evaluator. It successfully distinguishes between the quality tiers, with mean scores consistently following the expected Right > Safe > Wrong order for both captioning and QA tasks. Furthermore, its inter-rater reliability, measured by Fleiss' Kappa ($\kappa$), is substantial for QA ($\kappa = 0.73$) and moderate for captioning ($\kappa = 0.43$). However, the significant practical limitations of LLM-as-Judge method—including high computational cost, slow speed, and sensitivity to prompt engineering—motivate our development of the DATE metric as an efficient and scalable alternative.

*Table H.1.* Reliability of LLM-as-Judge (M.J.). Evaluation on three sets: **Right'** (rephrased references), **Safe'** (generic/vague domain responses), and **Wrong'** (incorrect random references). $\kappa$ denotes Fleiss' Kappa for inter-rater reliability; Mean' is the average M.J. score.

| Type | Mean | | Fleiss' Kappa ($\kappa$) | |
|---|---|---|---|---|
| | Caption | QA | Caption | QA |
| Right | 0.78 | 0.97 | 0.68 | 0.74 |
| Safe | 0.24 | - | 0.17 | - |
| Wrong | 0.13 | 0.12 | 0.45 | 0.72 |
| Overall | - | - | 0.43 | 0.73 |

# I. Task-specific prompts for LALM in MECAT tasks

This section details the prompt strategies employed for Large Audio-Language Models (LALMs) during the MECAT-Caption evaluation. For specific Audio-focused LALMs that require specialized instruction formats—namely Audio-Flamingo 2 (Ghosh et al., 2025) and Kimi-Audio (Ding et al., 2025)—the exact prompt templates are provided in Table I.1. For the remaining LALMs, the prompts are standardized as shown in Table I.2. This category encompasses a diverse range of architectures, including other Audio-focused models such as MiMo-Audio (Zhang et al., 2025), Step-Audio-2-mini (Wu et al., 2025), and Baichuan-Audio (Li et al., 2025); Omni LALMs represented by the Qwen-Omni series (Xu et al., 2025a;b; Qwen Team, 2025) and Baichuan-Omni (Li et al., 2024); Multimodal LALMs specifically Phi-4-Multimodal (Abouelenin et al., 2025); and the state-of-the-art Gemini series (Comanici et al., 2025; Google, 2025). Regarding system configurations, the system prompt was explicitly set to "*You are a helpful assistant*" for the Qwen2.5-Omni models, while the default system prompts were retained for all other architectures.

Regarding the MECAT-QA task, the prompt for each sample consists solely of the corresponding question, without additional task-specific templates.

*Table I.1.* Prompts for Audio-Flamingo2 and Kimi-Audio models in caption task

| Category | Subcategory | Prompt |
|---|---|---|
| Systematic | Short | Provide a caption for this audio within 15 words |
| | Long | Provide a caption for this audio within 1-2 sentences |
| Content-Specific | Speech | Provide a caption for the speech content in this audio |
| | Music | Provide a caption for the music content in this audio |
| | Sound | Provide a caption for general sound excluding speech and music |
| Content-Unrelated | Environment | Provide a caption for quality or acoustic environment for this audio |

*Table I.2.* Prompts for remaining models in caption task

| Category | Subcategory | Prompt |
|---|---|---|
| Systematic | Short | Listen to the audio and provide a caption for this audio within 15 words |
| | Long | Listen to this audio and provide a caption for this audio within 1-2 sentences |
| Content-Specific | Speech | Listen to the audio and provide a caption describing the speech content in this audio |
| | Music | Listen to the audio and provide a caption for the music content in this audio |
| | Sound | Listen to the audio and provide a general sound excluding speech and music |
| Content-Unrelated | Environment | Listen to this audio and provide a caption for quality or acoustic environment for this audio |

## J. Fine-grained Analysis of Speech Captioning

This section presents a detailed performance of all models on the pure speech subset of MECAT-Caption task. Table J.1 presents the results across three dimensions: similarity, discriminability, and DATE.

*Table J.1.* Comparison of different models. [†] indicates that its previous version (Audio Flamingo 2) was explicitly used in the data construction process.

| Type | Model | Similarity | | Discriminability | | DATE | |
|------|-------|------------|------|------------------|------|------|------|
| | | Score | Rank | Score | Rank | Score | Rank |
| Caption -Only | Pengi | 26.6 | 15 | 27.8 | 16 | 27.2 | 16 |
| | EnClap | 28.7 | 14 | 31.9 | 15 | 30.2 | 14 |
| LALM | Phi-4-Multimodal-Instruct | 26.6 | 16 | 27.2 | 17 | 26.9 | 17 |
| | Kimi-Audio-7B-Instruct | 25.6 | 17 | 36.2 | 14 | 30.0 | 15 |
| | Baichuan-Audio-Instruct | 37.2 | 10 | 60.3 | 7 | 46.0 | 8 |
| | Audio Flamingo 3[†] | 46.6 | 4 | 52.3 | 10 | 49.3 | 7 |
| | MiMo-Audio-Instruct | 42.5 | 7 | 49.7 | 11 | 45.8 | 9 |
| | Step-Audio-2-mini | 36.6 | 11 | 55.8 | 9 | 44.2 | 10 |
| | Baichuan-Omni | 34.9 | 13 | 57.7 | 8 | 43.5 | 11 |
| | Qwen2.5-Omni 3B | 37.3 | 9 | 49.4 | 12 | 42.5 | 12 |
| | Qwen2.5-Omni 7B | 35.3 | 12 | 45.9 | 13 | 39.9 | 13 |
| | Qwen3-Omni | 41.0 | 8 | 64.7 | 6 | 50.2 | 6 |
| | Qwen3-Omni-Flash-1201 | 47.7 | 2 | **80.8** | 1 | 59.2 | 3 |
| | Gemini-2.5-Flash | 45.8 | 5 | 77.2 | 5 | 57.5 | 4 |
| | Gemini-2.5-Pro | 44.3 | 6 | 78.4 | 4 | 56.6 | 5 |
| | Gemini-3-Flash | 47.5 | 3 | 79.3 | 3 | 59.4 | 2 |
| | Gemini-3-Pro | **48.8** | 1 | 79.6 | 2 | **60.5** | 1 |

## K. Model performance of similarity on MECAT Tasks

This section presents the complete Similarity scores for all models evaluated on MECAT, serving as a comparative reference for the DATE metrics reported in the main text (see Tables 2 and 3).

*Table K.1.* Model performance (Similarity %) on MECAT-Caption. **Bold** indicates the best performance, and underline indicates the second best. † indicates that its previous version (Audio Flamingo 2) was explicitly used in the data construction process.

| Type | Model | Systemic | | Content-Specific | | | | | | Content Unrelated | Score$_{Cap}$ |
| | | Long | Short | Speech | | Music | | Sound | | | |
| | | | | Pure | Mixed | Pure | Mixed | Pure | Mixed | Env | |
|---|---|---|---|---|---|---|---|---|---|---|---|
| Caption -Only | Pengi | 37.5 | 41.0 | 26.6 | 29.2 | 39.6 | 11.8 | 35.4 | 16.2 | 17.8 | 29.5 |
| | EnClap | 40.5 | 45.0 | 28.7 | 29.5 | 39.3 | 15.0 | 41.2 | 17.3 | 17.9 | 31.6 |
| LALM | Phi-4-Multimodal-Instruct | 45.4 | 40.3 | 26.6 | 31.7 | 41.5 | 26.2 | 29.5 | 25.7 | 37.3 | 37.4 |
| | Kimi-Audio-7B-Instruct | 40.8 | 45.7 | 25.6 | 27.1 | 39.5 | 16.2 | 35.8 | 19.4 | 16.7 | 30.8 |
| | Baichuan-Audio-Instruct | 33.0 | 28.2 | 37.2 | 35.0 | 36.4 | 24.7 | 45.0 | 29.9 | 47.1 | 36.1 |
| | Baichuan-Omni | 39.2 | 42.5 | 34.9 | 35.4 | 41.0 | 13.2 | 40.0 | 32.3 | 29.4 | 35.0 |
| | MiMo-Audio-Instruct | 49.9 | 49.4 | 42.5 | 43.5 | 47.5 | 19.9 | 44.5 | 27.6 | 27.2 | 41.2 |
| | Audio Flamingo 3† | 49.6 | 49.6 | 46.6 | 47.5 | 50.6 | 26.4 | 44.6 | 28.3 | 31.7 | 43.5 |
| | Qwen3-Omni | 38.2 | 33.6 | 34.1 | 34.5 | 49.0 | 34.1 | 41.4 | 20.8 | 40.2 | 37.4 |
| | Step-Audio-2-mini | 44.1 | 47.8 | 36.6 | 37.3 | 45.9 | 36.0 | 36.4 | 24.9 | 41.4 | 41.2 |
| | Qwen2.5-Omni 3B | 48.3 | 45.3 | 37.3 | 37.5 | 50.7 | 34.7 | 46.6 | 34.1 | 47.8 | 44.1 |
| | Qwen2.5-Omni 7B | 52.7 | 46.2 | 35.3 | 37.5 | 39.2 | 33.1 | 45.2 | 32.1 | 41.0 | 43.4 |
| | Qwen3-Omni-Flash-1201 | 53.6 | 50.4 | 46.7 | 47.6 | **61.8** | 38.2 | 50.4 | **36.5** | **52.7** | 50.7 |
| | Gemini-2.5-Flash | **56.1** | **53.5** | 45.8 | 46.6 | 59.1 | 44.3 | **50.7** | 36.4 | 48.9 | 51.0 |
| | Gemini-2.5-Pro | 50.8 | 49.9 | 44.3 | 45.7 | 58.5 | **44.6** | 49.6 | 35.0 | 51.9 | 49.3 |
| | Gemini-3-Flash | 53.1 | 50.8 | 47.5 | 49.2 | 53.6 | 42.6 | 49.8 | 36.0 | 50.8 | 50.3 |
| | Gemini-3-Pro | 53.2 | 53.3 | **48.8** | **50.7** | 58.0 | 43.1 | 50.5 | 34.9 | 49.4 | **53.1** |

*Table K.2.* Model Performance (Similarity %) on MECAT-QA. **Bold** indicates the best performance, and underline indicates the second best. [†] indicates that its previous version (Audio Flamingo 2) was explicitly used in the data construction process.

| Model | Perception | Analysis | | Reasoning | | | Score$_{QA}$ |
|---|---|---|---|---|---|---|---|
| | Direct Perception | Sound Characteristics | Quality Assessment | Environment Reasoning | Inference & Judgment | Application Context | |
| Kimi-Audio-7B-Instruct | 37.5 | 32.5 | 19.2 | 37.5 | 38.8 | 33.8 | 33.2 |
| Baichuan-Audio-Instruct | 35.2 | 36.6 | 36.0 | 38.1 | 39.5 | 39.6 | 37.5 |
| Baichuan-Omni | 36.8 | 36.1 | 35.4 | 39.1 | 38.5 | 39.4 | 37.6 |
| Phi-4-Multimodal-Instruct | 41.2 | 37.6 | 36.6 | 40.3 | 39.0 | 40.1 | 39.1 |
| MiMo-Audio-Instruct | 50.9 | 40.5 | 27.0 | 40.7 | 41.9 | 38.5 | 39.9 |
| Step-Audio-2-mini | 48.6 | 44.6 | 39.1 | 38.2 | 38.7 | 39.3 | 41.4 |
| Audio Flamingo 3[†] | 46.0 | 41.4 | 38.6 | 43.5 | 43.2 | 40.9 | 42.3 |
| Qwen2.5-Omni 3B | 47.2 | 43.8 | 39.7 | 43.2 | 41.0 | 41.9 | 42.8 |
| Qwen2.5-Omni 7B | 49.7 | 43.8 | 40.5 | 44.1 | 42.5 | 41.9 | 43.8 |
| Qwen3-Omni | **52.3** | 44.8 | **41.2** | 45.2 | 44.7 | 45.2 | 45.6 |
| Qwen3-Omni-Flash-1201 | 41.3 | 38.5 | 35.8 | 44.4 | 45.9 | 45.6 | 41.9 |
| Gemini-2.5-Flash | 47.9 | **46.1** | 39.7 | 46.2 | **47.1** | **47.9** | **45.8** |
| Gemini-2.5-Pro | 47.4 | 45.2 | 39.0 | **46.9** | 45.7 | 46.3 | 45.1 |
| Gemini-3-Flash | 46.0 | 42.3 | 34.0 | 44.7 | 45.4 | 46.1 | 43.1 |
| Gemini-3-Pro | 47.4 | 37.8 | 26.1 | 41.2 | 42.3 | 41.9 | 39.5 |

## L. Analysis of Speech Hallucination in Silent Segments

This section qualitatively evaluates the grounding capabilities of various models when presented with audio segments containing no discernible speech. We specifically analyze the "speech" output field for two representative silent cases (C1 and C2) to detect potential hallucinations in low-signal environments.

*Table L.1.* Speech caption generated by different models for two silence segments in MECAT. [†] indicates that its previous version (Audio Flamingo 2) was explicitly used in the data construction process.

| Type | Model | Output |
|------|-------|--------|
| Caption-Only | Pengi | **C1:** a person is playing a guitar.
**C2:** someone is opening and closing a door. |
| | EnClap | **C1:** A low frequency of a person talking in the distance
**C2:** A helicopter flying in the distance followed by a man speaking |
| LALM | Phi-4-Multimodal-Instruct | **C1:** Audio caption: A rumble is heard.
**C2:** Audio caption: A man is speaking with music in the background. |
| | Kimi-Audio-7B-Instruct | **C1:** a machine is running
**C2:** a person is playing a drum set with a bass drum and a snare drum. |
| | Baichuan-Audio-Instruct | **C1:** The speaker is talking about how they are going through their day... [Truncated]
**C2:** The speaker is asking for help with their website... [Truncated] |
| | Baichuan-Omni | **C1:** A woman is speaking with background noise.
**C2:** Someone is speaking with authority, likely giving instructions or delivering an important message. |
| | MiMo-Audio-Instruct | **C1:** A male voice, speaking in a neutral tone... says 'Thank you' in English...
**C2:** The audio contains a sequence of sound effects... with no speech. |
| | Audio Flamingo 3[†] | **C1:** A male voice says 'Thank you' amidst the sound of a waterfall.
**C2:** A female voice says 'Thank you' in a neutral tone. |
| | Qwen3-Omni | **C1:** A person is speaking.
**C2:** The audio contains only a single, sustained, low-pitched electronic tone. |
| | Step-audio-2-mini | **C1:** The speech content is "Oh no, I'm sorry."
**C2:** There is no speech in this audio. |
| | Qwen2.5-Omni 3B | **C1:** The audio contains a speech saying 'I'm gonna be a daddy'.
**C2:** The audio contains a speech segment where a male voice says 'you' in a neutral tone. |
| | Qwen2.5-Omni 7B | **C1:** The audio contains a speech segment where someone is saying 'I'm going to go ahead and do that.'
**C2:** The audio contains a speech segment in which the speaker says 'you'. |
| | Qwen3-Omni-Flash-1201 | **C1:** No speech detected in the audio.
**C2:** There is no speech in this audio. |
| | Gemini-2.5-Flash | **C1:** No discernible speech is present in this audio.
**C2:** No speech detected. |
| | Gemini-2.5-Pro | **C1:** There is no speech in this audio.
**C2:** There is no speech in this audio. |
| | Gemini-3-Flash | **C1:** A man provides a continuous spoken delivery, though the clarity is slightly affected by room acoustics.
**C2:** A male voice counts 'uno, dos, tres' in Spanish before the guitar playing begins. |
| | Gemini-3-Pro | **C1:** The audio does not contain any speech or vocalizations.
**C2:** A male voice speaks with an urgent and hushed tone, delivering a dramatic monologue. |

