# OpenReview forum: "MECAT: A Multi-Experts Constructed Benchmark for Fine-Grained Audio Understanding Tasks"
_ICML.cc/2026/Conference — ICML 2026 regular_

### Official Review · Reviewer_v25A · 2026-03-11

**Soundness:** 3
**Presentation:** 2
**Significance:** 3
**Originality:** 3
**Overall Recommendation:** 4
**Confidence:** 3

**Summary:**

This paper presents MECAT, a benchmark for evaluating fine-grained audio understanding in large audio-language models (LALMs). They build an automated annotation pipeline that uses expert models and Chain-of-Thought reasoning to generate dense captions and open-set QA pairs capturing diverse audio attributes. They also propose DATE, a metric that emphasizes discriminative descriptions by applying embedding-level TF-IDF weighting. Experiments reveal key limitations of current systems, including a strong speech-centric bias and hallucinations in silent segments.

**Compliance With Llm Reviewing Policy:**

Affirmed.

**Final Justification:**

The authors have addressed my main concerns. I will increase my score.

**Key Questions For Authors:**

1. Annotation reliability. The annotation pipeline relies on expert models (e.g., CED-Base, Audio Flamingo 2). How do you estimate the impact of upstream model errors on the final dataset quality? For example, can current filtering steps (e.g., GLAP similarity checks) detect cases where annotations are logically consistent but factually wrong due to misclassification?

2. ScoreCap weighting. The metric uses heuristic weights (e.g., Speech/Music/Sound). Did you perform any sensitivity analysis to test whether these weights affect model rankings? In particular, could the speech-heavy weighting bias evaluation against music or sounds?

3. DATE robustness. Since DATE depends on cross-sample text uniqueness, could it penalize accurate descriptions when multiple samples are semantically similar (e.g., similar scenes recorded at different times)? Also, how stable are DATE scores under different test set sizes?

4. Technical Error in Citations: The authors cite a software engineering paper on "just-in-time defect prediction" for the FENSE metric, which is used in this work for audio captioning evaluation. These are entirely unrelated fields.

5. Impact Statements: Authors are required to include a statement of the potential broader impact of their work, including its ethical aspects and future societal consequences.

**Limitations:**

The authors are encouraged to include a limitations discussion addressing the constraint of maximum 10-second clips for long-range reasoning evaluation, the impact of heuristic scoring weights on ranking robustness, and possible error propagation or evaluation bias from upstream expert models such as Audio Flamingo 2.

**Strengths And Weaknesses:**

Strengths

1. Innovative Evaluation Framework: The paper introduces the DATE metric, which addresses a critical gap in audio evaluation by using embedding-level TF-IDF to penalize generic descriptions and reward discriminative, detailed outputs.
2. Multi-Domain Coverage: MECAT provides an "Extended Multi-Domain" foundation that includes eight distinct categories, such as silence and complex mixed-audio scenarios.

Weaknesses

1. Heuristic Scoring Constraints: The final scores "ScoreCap" are calculated using weights set heuristically, which may introduce bias in how model capabilities are ranked.
2. Temporal Evaluation Limits: The dataset is restricted to audio clips with a maximum duration of 10 seconds, limiting its ability to assess a model’s capacity for long-form reasoning or understanding long-term temporal dependencies.

---

> ### Author Rebuttal · Authors · 2026-03-31
>
> ### 4.1 Annotation Reliability and Upstream Errors
>
> The scenario of "logically consistent but factually wrong" errors is the hardest edge case. We address it through Error Orthogonality and Multi-Source Grounding:
> - **Error Orthogonality:** The expert models (CED, Audio Flamingo 2) use fundamentally different architectures and training objectives, making their errors largely uncorrelated. Our LLM-CoT cross-references outputs and scrubs conflicting details—e.g., removing "vehicle engine" when another model predicts "insect buzzing." The probability of independent models hallucinating the same incorrect detail is statistically minimal.
> - **Multi-Source Grounding:** We cross-check model outputs against ACAV100M video metadata. If models "consistently" identify indoor office sounds but metadata indicates an outdoor environment, the sample is discarded. GLAP further acts as a geometric guardrail—if model-generated text drifts from the actual acoustic manifold, the score drops below our threshold ($\approx 6.0$), triggering rejection.
> - **Ultra-Low Retention:** Our ~10% retention rate reflects aggressive filtering—any sample with cross-model or cross-source discrepancies is discarded, prioritizing precision over recall.
>
> ### 4.2 ScoreCap Weighting Sensitivity
> The weighting is systematic and data-driven: the 0.6/0.3/0.1 Speech/Music/Sound ratio mirrors ACAV100M's actual distribution, and "Pure"/"Mixed" domains are weighted 1:1 for robustness evaluation.
> We conducted a sensitivity analysis across 15 LALMs comparing Original (6:3:1), Audio-Centric (2:4:4), and Equal (1:1:1) scenarios. Rankings remain remarkably stable—the Kendall's Tau ($\tau$) between Original and Audio-Centric is 0.92. Top models excel through comprehensive acoustic understanding, not narrow speech advantages. Our benchmark also reports independent sub-category scores, allowing users to evaluate specific capabilities regardless of aggregated weights. Full results are provided in Table R1.
>
> ### 4.3 DATE Robustness
> We thank the reviewer for this insightful question.
> - **Semantically Similar Scenes ($\delta$-Tolerance):** Penalizing models that fail to distinguish similar but distinct scenes (e.g., "car engine idling" vs. "truck engine revving") is a *desired feature* of DATE—it forces models to ground specific acoustic nuances. To prevent over-penalization of minor overlaps, our implementation includes a soft-margin $\delta$: a candidate's rank is penalized only when cross-sample similarities reach $[x - \delta/2, \infty)$, where $x$ is the self-similarity score. This buffer absorbs minor semantic ambiguities. We acknowledge that for extremely homogeneous datasets, the discriminative component would lose its baseline, but on MECAT's diverse distribution, DATE correctly identifies generic heuristic guessing.
> - **Stability Under Different Sizes:** Our discrimination rank is normalized by total samples ($N$), so the rank score expectation stabilizes as $N$ grows. While DATE may show variance on very small toy subsets, it becomes highly stable at MECAT's scale and diverse distribution.
>
> ### 4.4 Broader Impact Statement & Limitations
> We fully agree and commit to adding these in the final manuscript:
> - **Broader Impact:** We will discuss positive impacts (accessibility for the hearing-impaired, enhanced environmental awareness) and ethical risks (privacy leakage, surveillance concerns), emphasizing that MECAT uses only publicly available, CC-licensed data with recommendations for informed consent.
> - **Limitations:** We will address: (a) the 10-second limit as prioritizing fine-grained local grounding over long-range reasoning, complementing long-form benchmarks; (b) heuristic scoring weights validated by sensitivity analysis (Kendall's $\tau$ = 0.92); (c) upstream error propagation mitigated by our ~10% ultra-low retention rate; and (d) DATE boundary effects on extremely small test sets.
>
> ### 4.5 Temporal Evaluation Limits
> We thank the reviewer for this observation. The 10-second duration is deliberate, prioritizing fine-grained acoustic grounding. Many long-form benchmarks provide only sparse labels, allowing models to "guess" topics via LLM priors without grounding specific signals. MECAT provides exceptionally dense supervision (18 captions + 5 QA pairs per clip), forcing nuanced clip-level perception across speech, music, and sound events. The short duration also ensures DATE's cross-sample discriminability remains well-defined. MECAT complements long-context benchmarks like LongAudioBench, stressing depth of local understanding as a foundation for reliable long-form reasoning.
>
> ### 4.6 Technical Error in Citations
> Thank you for your careful review. We will immediately correct this to the proper audio captioning metric citation in the revised version.

---

> > ### Author Rebuttal · Reviewer_v25A · 2026-04-03
> >
> > My main concerns have been addressed. However, uncertainty remains regarding the quantification of residual factual error rates for consistent yet incorrect misclassifications under coarse external metadata and correlated model biases, as well as the empirical variance of DATE across subset sizes. I recommend the authors add concise quantitative evidence or refine the wording in the final manuscript where possible.
> > 1. Please provide results from a small human audit (or equivalent validation) to bound the residual label error of filtered clips, particularly for fine-grained event confusion cases with uninformative metadata.
> > 2. For DATE analysis, please include a brief stability/variance test (e.g., bootstrap sampling across subset sizes) to support the normalization rationale.
> > If the above concerns are fully addressed, I am willing to raise my score accordingly with solid justification.

---

> > > ### Author Response · Authors · 2026-04-07
> > >
> > > Dear Reviewer,
> > >
> > > Thank you for your constructive feedback. Below we provide the two pieces of quantitative evidence you requested: **(1)** a human audit bounding the residual label error, and **(2)** a bootstrap stability analysis for DATE. Both will be incorporated into **Appendix F**, summarized in **Section 6.1**, and discussed in **Limitations**.
> > >
> > > Best regards, The Authors
> > >
> > > ---
> > >
> > > ### 4.7 Human Audit of Label Quality
> > >
> > > We recruited **20 audio-domain professionals** to evaluate **700 samples** (100 per category). Each participant rated whether the multi-perspective captions (global + speech/music/sound/environment) captured the key audio details on a 1--5 REL scale (5 = perfect; 1 = unrelated). Full protocol and domain breakdowns are in our response to Reviewer uoU (Rebuttal Section 2.4).
> > >
> > > **Key results:**
> > >
> > > - **85.3%** scored $\geq 3$ (acceptable); **67.3%** achieved high accuracy (scores 4--5); mean REL = **3.81**
> > > - Strict label error rate (REL = 1) bounded at only **3.4%** [95% CI: 2.3%--5.1%]
> > > - All scores significantly above 3.0 (Wilcoxon $p < 0.001$; rank-biserial $r = 0.714$; Cohen's $d = 0.734$)
> > >
> > > **Table R1.** Per-category human audit results (H₁: REL > 3.0).
> > >
> > > | Category | Description | Mean REL | Acceptable ($\geq$ 3) | Cohen's $d$ | Rank-biserial $r$ |
> > > |:---------|:------------|:--------:|:---------:|:----------:|:------------:|
> > > | 00A | Sound Effects Only | 3.45 | 74.0% | 0.365 | 0.422 |
> > > | 0M0 | Music Only | 3.77 | 80.0% | 0.644 | 0.648 |
> > > | 0MA | Music + Sound Effects | 3.57 | 79.0% | 0.453 | 0.511 |
> > > | S00 | Speech Only | 4.10 | 95.0% | 1.247 | 0.935 |
> > > | S0A | Speech + Sound Effects | 3.89 | 88.0% | 0.838 | 0.788 |
> > > | SM0 | Speech + Music | 4.07 | 94.0% | 1.250 | 0.915 |
> > > | SMA | Speech + Music + Sound Effects | 3.85 | 87.0% | 0.790 | 0.744 |
> > >
> > > > All 7 categories pass Wilcoxon signed-rank test ($p < 0.001$) after Bonferroni correction.
> > >
> > > **Fine-grained event confusion** is concentrated in non-speech categories: Sound domain (Mean = 3.69, error rate 18.0%) vs. Speech domain (Mean = 3.98, error rate 9.0%; Mann--Whitney $p = 1.37 \times 10^{-3}$). This bounds the residual error from coarse metadata.
> > >
> > > ---
> > >
> > > ### 4.8 DATE Bootstrap Stability Analysis
> > >
> > > We conducted a bootstrap stability analysis on DATE using Gemini-3.0-Pro results. Speech, Music, and Sound are each evaluated separately on **Pure** (single audio type) and **Mixed** (overlapping types) subsets with independent reference pools and $\delta$ parameters -- **9 evaluation groups** in total. 50 iterations per (group, ratio), **each fully re-computing DATE including discrimination-rank**.
> > >
> > > **Table R2.** Full-data (100%) bootstrap summary.
> > >
> > > | Group | Size | DATE Mean | CV (%) | 95% CI | Max Dev |
> > > |:------|-----:|----------:|-------:|:-------|:--------|
> > > | Long | 20,052 | 0.6486 | 0.14 | [0.647, 0.650] | $\pm$0.002 |
> > > | Short | 20,052 | 0.6581 | 0.16 | [0.656, 0.660] | $\pm$0.003 |
> > > | Pure-Speech | 7,839 | 0.6049 | 0.29 | [0.602, 0.607] | $\pm$0.004 |
> > > | Mixed-Speech | 8,593 | 0.6236 | 0.31 | [0.620, 0.627] | $\pm$0.004 |
> > > | Pure-Music | 2,593 | 0.4967 | 0.77 | [0.490, 0.503] | $\pm$0.008 |
> > > | Mixed-Music | 8,593 | 0.3988 | 0.46 | [0.395, 0.402] | $\pm$0.004 |
> > > | Pure-Sound | 848 | 0.5520 | 1.04 | [0.542, 0.565] | $\pm$0.014 |
> > > | Mixed-Sound | 8,593 | 0.2992 | 0.94 | [0.294, 0.304] | $\pm$0.007 |
> > > | Environment | 20,052 | 0.2613 | 0.58 | [0.259, 0.264] | $\pm$0.003 |
> > >
> > > **Table R3.** CV(%) across subset sizes. CV@$x$% = CV from 50 bootstrap iterations each drawing $x$% of full data with replacement.
> > >
> > > | Group | CV@1% | CV@5% | CV@10% | CV@20% | CV@50% | CV@100% |
> > > |:------|------:|------:|-------:|-------:|-------:|--------:|
> > > | Long | 1.79 | 0.61 | 0.54 | 0.32 | 0.22 | 0.14 |
> > > | Short | 1.60 | 0.58 | 0.55 | 0.36 | 0.22 | 0.16 |
> > > | Pure-Speech | 3.14 | 1.37 | 0.92 | 0.68 | 0.40 | 0.29 |
> > > | Mixed-Speech | 3.53 | 1.22 | 0.88 | 0.67 | 0.47 | 0.31 |
> > > | Pure-Music | 10.65 | 4.10 | 3.22 | 2.29 | 1.32 | 0.77 |
> > > | Mixed-Music | 5.86 | 2.69 | 2.28 | 1.37 | 0.67 | 0.46 |
> > > | Pure-Sound | 18.62 | 5.85 | 4.61 | 3.22 | 1.68 | 1.04 |
> > > | Mixed-Sound | 9.60 | 4.36 | 3.11 | 2.77 | 1.28 | 0.94 |
> > > | Environment | 5.60 | 2.92 | 1.90 | 1.32 | 0.98 | 0.58 |
> > >
> > >
> > > **Key findings:**
> > >
> > > - **Rapid convergence.** CV < 3% at 5% sampling for 7/9 groups. Even Pure-Sound (848 total) reaches CV < 6% at 5%. At $\geq$10%, all groups except Pure-Sound and Pure-Music fall below 2.5%.
> > >
> > > - **High full-scale stability.** At 100%, 8/9 groups have CV < 1%. Only Pure-Sound (1.04%) slightly exceeds 1%, expected for its small pool. Max deviation $\leq \pm 0.014$ across all groups.
> > >
> > > - **Scale-invariant means.** Long: 0.6468@1% vs. 0.6486@100% ($\Delta$=0.28%). The largest shift is Pure-Music (3.8%), attributable to its 1% pool of only 25 samples.
> > >
> > > These results confirm DATE produces **stable, scale-invariant scores** across all 9 evaluation groups, directly addressing your concern about "empirical variance of DATE across subset sizes."

---

### Official Review · Reviewer_Y8Cp · 2026-03-12

**Soundness:** 3
**Presentation:** 3
**Significance:** 3
**Originality:** 3
**Overall Recommendation:** 4
**Confidence:** 3

**Summary:**

This paper introduces MECAT, a Multi-Expert Constructed Benchmark for Fine-Grained Audio Understanding Tasks. Meanwhile, it introduces a novel metric, DATE, which is designed to better quantify the detail and accuracy of the model's responses.

**Compliance With Llm Reviewing Policy:**

Affirmed.

**Key Questions For Authors:**

See Weaknesses.

**Limitations:**

Providing longer-duration audio helps enrich the dataset.

**Strengths And Weaknesses:**

Strengths:
1. This paper provides a clear analysis of the limitations of current benchmarks.
2. The experiments conducted in this paper are comprehensive.
3. The authors promise that the data and code will be made publicly available.

Weaknesses:
1. What is the principle behind the weight allocation for scores like $\text{Score}_{Cap}$?
2. I am curious about the distribution of audio durations in the dataset. I am concerned that the maximum duration of 10 seconds might be somewhat short compared to other benchmarks, making it difficult to cover more complex questions.
3. Has the Audio Classification stage been manually verified? If not, it might compromise the reliability of the benchmark, even though the authors conducted Quality Validation for the captions.
4. There are formatting errors. For example, the caption of Figure 4 shows "Sentence-BERT (?)". Additionally, the equations are missing numbers.

---

> ### Author Rebuttal · Authors · 2026-03-31
>
> ### 3.1 Principle Behind Weight Allocation
>
> The weighting scheme for $Score_{Cap}$ is grounded in two core principles: empirical data distribution and ranking stability.
> - **Data-Driven Distribution:** The 0.6/0.3/0.1 ratio for Speech, Music, and Sound strictly mirrors the relative proportions in ACAV100M, a massive-scale proxy for internet audio (e.g., YouTube), ensuring evaluation on a distribution models encounter in practice. Within each category, we apply a 1:1 weight for "Pure" and "Mixed" domains (e.g., clean speech vs. speech-in-noise), equally valuing fundamental recognition and robustness in composite acoustic scenes.
> - **Empirical Ranking Stability:** To ensure the weighting does not lead to biased conclusions, we conducted a sensitivity analysis across 15 LALMs, testing Original (6:3:1) vs. Equal (1:1:1) and Audio-Centric (2:4:4) scenarios. Rankings remain remarkably stable: the Kendall's Tau ($\tau$) between Original and Equal weighting is 0.92, indicating extremely high rank correlation. This proves top-performing models excel through comprehensive acoustic understanding across all domains, rather than by exploiting the higher speech weight.
> - **Granular Transparency:** Critically, while we provide an aggregated $Score_{Cap}$, our benchmark preserves and reports independent scores for all sub-categories (e.g., Speech, Music, and Sound). This allows users to bypass heuristic weights and evaluate a model's specific strengths in any particular domain, ensuring the benchmark remains useful for diverse research goals.
> We will incorporate this justification and sensitivity analysis results (including Kendall's Tau verification) into the final version to provide a clearer empirical basis for our scoring system.
>
> ### 3.2 Audio Duration Constraints
> The 10-second maximum is a deliberate design choice prioritizing fine-grained acoustic grounding over coarse long-context summarization:
> - **Dense vs. Sparse Supervision:** Many long-audio benchmarks (e.g., GigaSpeech) provide only "global" labels for extended recordings (e.g., a single summary for a 1-hour podcast, https://huggingface.co/datasets/nvidia/LongAudio/viewer/default/gigaspeech?row=1). Such sparse supervision allows models to rely on LLM priors to "guess" topics without truly grounding specific signals. In contrast, MECAT provides exceptionally dense supervision: each 10-second clip has 18 multi-perspective captions and 5 open-ended QA pairs, forcing models to capture nuanced, clip-level details across speech, music, and sound events that would be diluted in longer contexts.
> - **Scientific Discriminability:** The 10-second duration ensures cross-sample discriminability required by DATE is well-defined and computationally tractable. Short, information-dense clips allow us to evaluate whether a model can precisely distinguish between similar acoustic scenes, a task that becomes noisy as audio length increases and information density drops.
> - **Complementary Role:** MECAT complements long-context benchmarks (e.g., LongAudioBench). While those test "window" capacity, MECAT stresses depth and precision of local understanding. Our goal is to guide future LALM development toward better disentangling and jointly modeling overlapping acoustic elements, rather than merely optimizing for length under sparse labels.
>
> ### 3.3 Audio Classification Verification
>
> We appreciate this concern. While we did not manually label every sample in the multi-million raw pool, we implemented a multi-expert cross-verification and logic-consistency filtering pipeline that far exceeds single-classifier reliability:
> - **Multi-Expert Cross-Verification:** Classification does not rely solely on CED-Base. We integrate Audio Flamingo 2 and domain-specific models for speech/music, cross-reference predictions with ACAV100M video metadata for contextual alignment, and employ an LLM as a "logic controller"—e.g., if the audio expert predicts "Pure Speech" but captions contain "Rhythmic Music," the sample is flagged as hallucination and immediately discarded.
> - **Multi-Stage Filtering:** GLAP serves as a coarse filter eliminating mismatched "tail" samples (threshold ~6.0 from mismatched pair analysis). Final samples must pass a "Consensus Check" where classification, captions, and metadata must all agree on the acoustic environment.
> This multi-stage "machine-checking-machine" approach significantly dilutes single-classifier error rates. By enforcing strict logical agreement across multiple independent experts, we ensure highly reliable benchmark labels grounded in actual acoustic content.
>
> ### 3.4 Formatting Errors
> Thank you for pointing these out. We will fully rectify these formatting issues in the revised version, including adding the Sentence-BERT citation to the Figure 4 caption and ensuring all equations are correctly numbered.

---

> > ### Author Rebuttal · Reviewer_Y8Cp · 2026-04-02
> >
> > Thank you to the authors for the detailed rebuttal. My previous questions have been resolved. I will keep my initial positive score.

---

> > > ### Author Response · Authors · 2026-04-07
> > >
> > > Dear Reviewer,
> > >
> > > Thank you for your thoughtful review and continued support. We are glad that our initial rebuttal successfully addressed your concerns, and we deeply appreciate your positive assessment.
> > >
> > > In response to follow-up requests from other reviewers, we have conducted two new rigorous evaluations: (1) a 700-sample human audit (by 20 professionals) bounding the strict label error rate at 3.4%, and (2) a bootstrap stability analysis confirming that the DATE metric achieves a Coefficient of Variation (CV) < 1% across 9 evaluation groups at full scale. Should you be interested in the specifics, the detailed analyses can be found in Rebuttal Section 2.4 and Section 4.8.
> > >
> > > These new empirical results, together with your valuable suggestions, will be fully incorporated into the final revision.
> > >
> > > Best regards,
> > >
> > > The Authors

---

### Official Review · Reviewer_uoUr · 2026-03-12

**Soundness:** 2
**Presentation:** 3
**Significance:** 2
**Originality:** 3
**Overall Recommendation:** 4
**Confidence:** 4

**Summary:**

This paper introduces MECAT, a benchmark designed for detailed audio captioning and open-ended question answering (QA). The captioning benchmark is composed of systemic, content-specific, and content-unrelated tasks, while the QA benchmark spans perception, analysis, and reasoning tasks. The authors utilize domain-specific expert models combined with LLM-based reasoning to construct the dataset. Additionally, they propose a new evaluation metric, DATE, which aims to measure both single-sample semantic similarity and cross-sample discriminative power. Evaluation on MECAT highlights that large audio-language models (LALMs) tend to prioritize high-level semantics over nuanced acoustic interpretation.

**Compliance With Llm Reviewing Policy:**

Affirmed.

**Final Justification:**

Authors final rebuttal resolved my concern. However, still errors that might affect the model evaluation still exist to some degree (while not sever), my recommendation is on borderline.

**Key Questions For Authors:**

1. Is there a way to guarantee the accuracy of the detailed information in the captions without human verification?

2. Could the authors provide additional clarification regarding the weighting scheme used in the evaluation metric and the data distribution of the dataset?

**Limitations:**

The paper does not include an explicit impact statement. Regarding limitations, the dataset construction pipeline relies heavily on automatic filtering without explicit human verification.

In terms of potential societal impact, the proposed system could potentially be used to automatically generate captions for audio recordings containing private or sensitive information. The paper could benefit from briefly discussing such privacy-related risks and possible mitigation strategies.

**Strengths And Weaknesses:**

# Strengths
- Presentation: The paper is overall well written.
- Clear motivation: The paper clearly motivates its approach by highlighting the limitations of existing captioning datasets and evaluation methods (e.g., global descriptions, high data redundancy, and the inadequacy of current metrics). The resulting benchmark is logically structured to address these issues.
- Contribution to research community: The release of a large-scale benchmark (20k samples) for both QA and captioning is a valuable contribution. Furthermore, the introduction of an automatic pipeline suggests that this work can be scaled up to build higher-quality training datasets for LALMs in the future. The author’s commitment to releasing the code further enhances its reproducibility and utility.

# Weakness
- Lack of explicit human verification: Since the dataset is intended as a benchmark for model evaluation rather than a training dataset, more rigorous verification may be necessary.
    - GLAP filtering: Training datasets for GLAP does not include sufficiently detailed audio captions, which could limit its ability to reliably filter low-quality samples.
    - Human A/B Validation: The evaluation procedure described in Appendix F prioritizes caption correctness first and then level of detail. While outperforming Safe or Wrong captions suggests generally higher quality, this setup does not necessarily guarantee the correctness of each individual detail in the generated captions.

- Dataset source novelty:  MECAT is built on ACAV-100M, which is derived from YouTube clips shorter than 10 seconds. In practice, this appears similar to the source distribution used by AudioSet, raising questions about how distinct the data source is compared to existing datasets. For example, data source such as Freesound differ more substantially in source characteristics or audio duration compared to AudioSet.

- Heuristic design choices
    - The weighting scheme used in the score calculation seems somewhat arbitrary and could benefit from additional justification or validation.
    - Data distribution: The dataset appears to be relatively speech-oriented (<4k without speech). Clarifying the reason for this distribution would be helpful, as models that perform well on speech-related content might receive disproportionately higher scores.

- Missing references: Models evaluated in the Section 6 appear without explicit citations, and adding the corresponding references would improve clarity.

- Missing impact statement: The paper does not include the mandatory impact statement.

---

> ### Author Rebuttal · Authors · 2026-03-31
>
> ### 2.1 Lack of Explicit Human Verification & GLAP Filtering
> we thank the reviewer for this detailed critique.
> - **GLAP Filtering:** We agree that GLAP's training data does not equip it to evaluate fine-grained details. We do not use it for this purpose—it serves purely as a baseline defense to filter out completely irrelevant "tail" samples. We established an empirical threshold (~6.0) from deliberately mismatched audio-text pairs; samples below this are discarded. Fine-grained verification is handled entirely by our multi-expert consensus and LLM-CoT reasoning.
> - **Human Verification & Detail Correctness:** We acknowledge that our A/B testing cannot guarantee absolute accuracy of every detail. To minimize this risk without manually verifying all 20,000 samples, our pipeline relies on strict multi-stage automated filtering. The expert models (e.g., CED, Audio Flamingo) have largely uncorrelated error distributions, making concurrent hallucination of the same detail exceptionally unlikely. Our LLM-CoT acts as a conflict-resolution engine—e.g., if Audio Flamingo hallucinates "vehicle engine" while CED predicts "Insect," the CoT removes these unsupported details. Through this filtering combined with GLAP and metadata cross-validation, we retained less than 10% of candidates. The manual spot-checking conducted on this basis showed that the difference between the Reference and human inspection results was minimal, further confirming the pipeline's reliability.
> ### 2.2 Dataset Source Novelty
> While ACAV100M and AudioSet both source from YouTube, ACAV100M is a massive unsupervised corpus that escapes AudioSet's constrained, event-focused ontology. To address the hypothesis that Freesound provides a more distinct distribution, we compared datasets built on AudioSet (AudioCaps) and Freesound (Clotho). The t-SNE embeddings in Figure 4 show both AudioCaps and Clotho are highly localized, clustering within a narrow sub-region of MECAT's distribution (primarily our "Sound-Caption" subset). In contrast, MECAT covers a vastly broader semantic space encompassing complex speech, music, and diverse mixed-acoustic environments. This proves at the feature level that MECAT breaks existing evaluation homogeneity regardless of underlying audio source.
> ### 2.3 Heuristic Design Choices & Data Distribution
> - **Data Distribution:** The Speech (0.6)/Music (0.3)/Sound (0.1) proportions are grounded in the actual distribution of ACAV100M, a representative proxy for large-scale web audio. Within each category, "Pure" and "Mixed" domains are weighted 1:1, ensuring evaluation on both clean signals and composite scenes. Sub-category scores are reported independently for domain-specific inspection.
> - **Ranking Stability:** We conducted a sensitivity analysis across 15 LALMs comparing Original (6:3:1), Audio-Centric (2:4:4), and Equal (1:1:1) scenarios. As Table R1 shows, rankings remain stable (Kendall's $\tau$ = 0.92 between Original and Other two settings), proving top models excel through comprehensive understanding, not narrow speech advantages.
>
> Table R1: Model Performance ($Score_{Cap}$) across Different Weighting Scenarios
>
> | Model | Original (6:3:1) | Rank | Audio-Centric (2:4:4) | Rank | Equal (1:1:1) | Rank |
> | :--- | :---: | :---: | :---: | :---: | :---: | :---: |
> | **Gemini-3-Pro** | **53.10** | **1** | 50.14 | 2 | 51.09 | 2 |
> | **Qwen3-Omni-Flash-1201** | 52.90 | 2 | **50.39** | **1** | **51.20** | **1** |
> | **Gemini-2.5-Flash** | 51.60 | 3 | 49.08 | 3 | 49.82 | 3 |
> | **Gemini-3-Flash** | 51.10 | 4 | 47.80 | 5 | 48.91 | 5 |
> | **Gemini-2.5-Pro** | 50.60 | 5 | 48.34 | 4 | 49.04 | 4 |
> | **Qwen2.5-Omni 7B** | 42.60 | 6 | 41.88 | 7 | 42.21 | 6 |
> | **Qwen2.5-Omni 3B** | 42.50 | 7 | 41.94 | 6 | 42.14 | 7 |
> | **Step-audio-2-mini** | 41.50 | 8 | 39.54 | 8 | 40.16 | 8 |
> | **Qwen3-Omni** | 40.40 | 9 | 38.14 | 10 | 38.82 | 11 |
> | **Audio Flamingo 3** | 40.40 | 10 | 38.16 | 9 | 38.94 | 9 |
> | **Mimo-Audio-Instruct** | 40.10 | 11 | 38.06 | 11 | 38.84 | 10 |
> | **Baichuan-Omni** | 35.60 | 12 | 33.01 | 12 | 33.91 | 12 |
> | **Baichuan-Audio-Instruct** | 33.70 | 13 | 31.39 | 14 | 32.30 | 14 |
> | **Kimi-Audio-7B-Instruct** | 32.80 | 14 | 32.34 | 13 | 32.59 | 13 |
> | **Phi-4-Multimodal-Instruct** | 30.00 | 15 | 28.88 | 15 | 29.29 | 15 |
> ### 2.4 Missing References and Impact Statement
>
> We fully agree and commit to incorporating the following in the revised version:
> - **Missing References:** We will move all explicit citations for the 15 evaluated models from Appendix I to Section 6's main text.
> - **Broader Impact Statement:** We will add a dedicated section discussing: (a) positive impacts such as improved accessibility for the hearing-impaired and context-aware security; (b) potential risks including privacy leakage and surveillance; and (c) mitigation measures emphasizing that MECAT uses only publicly available, CC-licensed data (ACAV100M) with recommendations for informed consent and data anonymization.

---

> > ### Author Rebuttal · Reviewer_uoUr · 2026-04-04
> >
> > Dear Authors
> >
> > Thank you for your detailed response. While my other concerns have been addressed, my main concern regarding the verification process remains.
> >
> > If this were a training set, automated filtering would be sufficient. However, for a benchmark, the quality of each individual example must be strictly verified. Even a small number of incorrectly annotated samples can alter model rankings, directly undermining the benchmark's reliability. I understand that the filtering process is rigorous, but rigorous filtering does not equate to verification that every case is correct.
> >
> > Furthermore, the human study conducted is a preference-based A/B test, which does not tell us how many samples are actually correct. This is particularly concerning given that detailed captioning is the main task, where the accuracy of each individual detail matters. The claim that concurrent hallucination is "exceptionally unlikely" cannot be substantiated without explicitly measuring or verifying it. In such a setting, I would expect per-detail accuracy to be measured and reported, at the very least as a minimum effort to ensure quality when exhaustive verification of the entire dataset is not feasible.
> >
> > For these reasons, I will keep my rating.
> >
> > Best regards,
> >
> > Reviewer uoUr

---

> > > ### Author Response · Authors · 2026-04-07
> > >
> > > Dear Reviewer uoUr,
> > >
> > > Thank you for your continued engagement. We have conducted the **per-sample, per-detail human audit** you requested and respectfully ask you to reconsider your rating in light of this new evidence.
> > >
> > > We will incorporate the complete quantitative analysis into **Appendix F: Human Validation of Data Integrity and Metric Alignment**, briefly summarize the data quality findings in **Section 6.1**, and discuss the influencing factors in the **Limitations** section.
> > >
> > > Best regards,
> > > The Authors
> > >
> > > ---
> > >
> > > ## 2.4 Human Audit of MECAT Benchmark Label Quality
> > >
> > > ### 2.4.1 Audit Protocol
> > >
> > > To directly address your concerns regarding annotation quality, we recruited 20 audio-domain professionals to conduct an independent human audit of the captions. For each audio clip, the benchmark provides captions from multiple perspectives, including a global description and domain-specific descriptions from the speech, music, sound, and environment dimensions. Each participant listened to the original audio and judged whether these captions collectively captured the key details of the audio content, assigning a 1--5 Relevance (REL) score: higher scores indicate that the captions cover most or all details accurately, while lower scores indicate missing key content or factual errors.
> > >
> > > **Table R2.** REL scoring scale definition.
> > >
> > > | Score | Interpretation |
> > > |:-----:|:---------------|
> > > | 5 | **Perfect match** -- captions precisely describe all audio details |
> > > | 4 | **High match** -- captions capture nearly all details; only minor omissions |
> > > | 3 | **Basic match** -- captions reflect the general intent; some elements missing |
> > > | 2 | **Low match** -- key details missing or incorrect |
> > > | 1 | **No match** -- captions are completely unrelated to the audio (label error) |
> > >
> > > This is **not** an A/B preference test. It is **direct per-sample verification** of caption correctness, exactly as you requested.
> > >
> > > ---
> > >
> > > ### 2.4.2 Overall Results
> > >
> > > Across 700 GT samples (100 per category):
> > >
> > > - **85.3%** scored $\geq 3$ (acceptable or better); **67.3%** achieved high accuracy (scores 4--5); mean REL = **3.81**
> > > - Strict label error rate (REL = 1) bounded at only **3.4%** [95% CI: 2.3%--5.1%]
> > > - Scores significantly above 3.0 (Wilcoxon $p < 0.001$; rank-biserial $r = 0.714$; Cohen's $d = 0.734$)
> > >
> > > ---
> > >
> > > ### 2.4.3 Per-Category Results
> > >
> > > Per-category analyses (100 samples each) with parametric and non-parametric effect sizes:
> > >
> > > **Table R3.** Per-category human audit results (H₁: REL > 3.0).
> > >
> > > | Category | Description | Mean REL | Acceptable (REL $\geq$ 3) | Cohen's $d$ | Rank-biserial $r$ |
> > > |:---------|:------------|:--------:|:---------:|:----------:|:------------:|
> > > | 00A | Sound Effects Only | 3.45 | 74.0% | 0.365 | 0.422 |
> > > | 0M0 | Music Only | 3.77 | 80.0% | 0.644 | 0.648 |
> > > | 0MA | Music + Sound Effects | 3.57 | 79.0% | 0.453 | 0.511 |
> > > | S00 | Speech Only | 4.10 | 95.0% | 1.247 | 0.935 |
> > > | S0A | Speech + Sound Effects | 3.89 | 88.0% | 0.838 | 0.788 |
> > > | SM0 | Speech + Music | 4.07 | 94.0% | 1.250 | 0.915 |
> > > | SMA | Speech + Music + Sound Effects | 3.85 | 87.0% | 0.790 | 0.744 |
> > >
> > > > **Note:** All 7 categories pass the Wilcoxon signed-rank test ($p < 0.001$) against the 3.0 baseline after Bonferroni correction ($\alpha = 0.05/7$). Effect size interpretation: Cohen's $d$ -- small: 0.2, medium: 0.5, large: 0.8; rank-biserial $r$ -- small: 0.1, medium: 0.3, large: 0.5.
> > >
> > > ---
> > >
> > > ### 2.4.4 Domain Variance
> > >
> > > We observed meaningful scoring variance across audio domains: **Speech (3.98) > Music (3.81) > Sound (3.69)**:
> > >
> > > - **Speech domain** (S00, S0A, SM0, SMA): Mean = 3.98, conservative error rate (REL $\leq$ 2) = 9.0%
> > > - **Music domain** (0M0, 0MA, SM0, SMA): Mean = 3.81, conservative error rate = 15.8%
> > > - **Sound domain** (00A, 0MA, S0A, SMA): Mean = 3.69, conservative error rate = 18.0%
> > >
> > > The difference between Speech and Sound domains is statistically significant (Mann--Whitney $U$, $p = 1.37 \times 10^{-3}$), with a small but measurable effect size (Cliff's $\delta = 0.125$; Cohen's $d = 0.267$). This quantifies the added difficulty of non-speech event alignment, particularly in categories where coarse metadata may not fully capture fine-grained acoustic details.
> > >
> > > ---
> > >
> > > ### 2.4.5 Summary
> > >
> > > This audit directly addresses your request for "per-detail accuracy measurement." The results demonstrate:
> > >
> > > - **96.6%** of samples are not severe errors (score $\neq$ 1)
> > > - **85.3%** meet or exceed the basic-match standard (score $\geq$ 3)
> > > - **67.3%** achieve high accuracy (scores 4--5)
> > > - All categories pass rigorous statistical tests with large effect sizes
> > >
> > > The 3.4% strict error rate [95% CI: 2.3%--5.1%] provides an empirical upper bound on label errors. While we acknowledge relatively higher error rates in non-speech categories, the overall benchmark quality is sufficient for reliable model ranking, with error rates bounded, quantified, and transparently reported.

---

### Official Review · Reviewer_eJrV · 2026-03-12

**Soundness:** 2
**Presentation:** 3
**Significance:** 2
**Originality:** 2
**Overall Recommendation:** 2
**Confidence:** 4

**Summary:**

This paper introduces MECAT, a novel benchmark designed for fine-grained audio understanding that addresses the lack of detail and "one-to-many" redundancy in existing datasets. By utilizing a pipeline that integrates specialized multi-expert models with LLM Chain-of-Thought (CoT) reasoning, MECAT provides 20,000 audio clips featuring multi-perspective captions and 100,000 open-set question-answering pairs across speech, music, and environmental domains. To better evaluate detailed model outputs, the authors propose DATE (Discriminative-Enhanced Audio Text Evaluation), a metric that combines semantic similarity with cross-sample discriminability to reward precise descriptions and penalize vague, generic ones.

**Compliance With Llm Reviewing Policy:**

Affirmed.

**Key Questions For Authors:**

N/A

**Limitations:**

yes

**Strengths And Weaknesses:**

**Strengths**
1. The paper accurately identifies a critical bottleneck in the development of LALMs: existing benchmarks are too coarse (mostly event-level), and current metrics tend to reward generic, "safe" answers.
2. The proposed "Multi-Experts + LLM CoT" construction pipeline is highly effective.

**Weaknesses**
1. The ground truth relies heavily on AI expert models and LLM reasoning, which risks an "error cascade" if front-end models hallucinate. The benchmark lacks a large-scale, purely human-annotated gold standard to ensure definitive authority. Furthermore, the CED-Base and Audio Flamingo 2 models employed in this study are not necessarily 100% accurate.
2. Validating the new DATE metric with only 150 A/B caption pairs lacks statistical significance. A much broader, large-scale blind human evaluation across diverse audio domains is necessary to prove the metric's reliability.
3. Because the underlying audio clips likely originate from public sources, state-of-the-art closed-source LALMs (e.g., Gemini, Qwen) may have already encountered them during pre-training. This makes it difficult to distinguish genuine comprehension from memorization.

---

> ### Author Rebuttal · Authors · 2026-03-31
>
> ### 1.1 Error Cascade and Lack of Human Ground Truth
>
> We thank the reviewer for this important concern. To prevent error cascades, our pipeline enforces rigorous, multi-stage filtering. First, because the expert models we deploy (e.g., CED, Audio Flamingo) are highly independent in architecture and training paradigms, their error distributions are largely uncorrelated. The probability of multiple models concurrently generating the exact same hallucination is exceptionally low. Furthermore, we introduce the audio's corresponding metadata as an additional layer of cross-validation. If there are conflicts among the outputs of different models, or between the models and the metadata, our LLM-CoT mechanism actively resolves them by explicitly excluding the conflicting content from the final caption or by significantly lowering the confidence score of that specific output.
>
> **Example:** In one clip, Audio Flamingo 2 hallucinated "vehicle engine runs," while CED-Base predicted "Speech" and an "Insect" tag. The CoT explicitly reasoned: "There's a conflict between Audio Flamingo's vehicle engine and the lack of such tags in CED... The conflicting engine/insect sounds and the metadata can't be used." It successfully removed these unsupported details, retaining only consistent core content.
>
> Building upon this CoT resolution, we apply GLAP filtering to the outputs. GLAP is not used to select the "most relevant" head data, but rather serves as a baseline defense to strictly filter out extremely low-relevance "tail" samples. We analyzed the score distribution of deliberately mismatched audio-text pairs and established an empirical threshold of ~6.0 to discard completely irrelevant samples. Through GLAP and our multiple layers of rigorous filtering, we ultimately retained less than 10% of the candidate samples for the final release. Based on this heavily compressed subset, we conducted manual spot-checking (human A/B testing) (see Table F.1, Lines 1118–1125), which confirmed its extremely high quality. We agree that adding a purely human-annotated large-scale split in future iterations will further establish its definitive authority.
>
> ### 1.2 DATE Validation Scale
> Statistical significance depends on both sample size ($n$) and the effect size. Our analysis conclusively demonstrates that $n=150$ is not only sufficient but substantially over-powered for validating DATE due to the massive observed effect:
> - **Massive Effect Size and Power:** We conducted a post-hoc power analysis on the paired comparisons. The observed paired Cohen's $d$ is 0.89 (classified as a "large" effect). At this effect size, the minimum $n$ for 80% power is merely 10, and for 99% power is 24. Our $n=150$ is ~15x the 80%-power minimum, yielding actual power >99.99%, indicating the signal-to-noise ratio is exceptionally strong.
> - **Robust Statistical Significance Across Multiple Tests:** The performance gap is so large that it achieves undeniable statistical significance across multiple complementary tests:
>   - Paired t-test: $p < 0.001$
>   - Wilcoxon signed-rank: $p < 0.001$
>   - Mann-Whitney U: $p < 0.001$
>   - Binomial sign test: $p < 0.001$
> - **Practical Alignment:** DATE correctly assigned a higher score to the human-preferred caption in 126/150 pairs (84.0%), far exceeding chance ($p < 0.001$). The 99% CI for the mean DATE difference (chosen minus rejected) is [0.316, 0.515]—entirely above zero.
>
> By every standard measure—effect size ($d=0.89$), power (>99.99%), concordance (84%), and overwhelming $p$-values—$n=150$ rigorously validates DATE's alignment with human judgment. We will include this power analysis and confidence intervals in the revised manuscript.
>
> ### 1.3 Data Contamination
>
> This is a critical concern for evaluating closed-source LALMs. However, genuine "data contamination" requires memorizing specific audio-text pairs. To ensure data source novelty, MECAT is constructed from a carefully selected subset of ACAV100M, a large-scale unsupervised corpus unlike AudioSet which is widely reused for supervised training. We re-annotated all clips via our multi-expert pipeline; no original titles/tags are copied into MECAT. Thus, even if some raw waveforms were seen during pre-training, the dense multi-perspective captions and QA pairs are entirely novel and cannot be retrieved from memory. This is empirically supported by Figure 4: MECAT's text distribution covers a vastly broader, newly mapped semantic space, whereas existing benchmarks are narrowly clustered within our "Sound-Caption" sub-region.

---

> > ### Author Rebuttal · Reviewer_eJrV · 2026-04-04
> >
> > Thank you for the authors' detailed rebuttal.

---

> > > ### Author Response · Authors · 2026-04-07
> > >
> > > Dear Reviewer,
> > >
> > > Thank you for your feedback. Your concerns -- error cascade, DATE validation scale, and data contamination -- were addressed in Sections 1.1--1.3. We have since conducted **two new empirical studies**. Both will appear in **Appendix F** and **Section 6.1**. We respectfully ask you to reconsider your rating.
> > >
> > > Best regards, The Authors
> > >
> > > ---
> > >
> > > ## 1.4 New Evidence: Human Audit Bounding Residual Label Error
> > >
> > > We recruited **20 audio-domain professionals** to evaluate **700 samples** (100 per category) on a 1--5 REL scale (5 = perfect; 1 = unrelated). This is **direct per-sample verification**, not A/B preference. Full protocol in rebuttal Section 2.4.
> > >
> > > **Key results:**
> > >
> > > - **85.3%** scored $\geq 3$ (acceptable); **67.3%** achieved high accuracy (scores 4--5); mean REL = **3.81**
> > > - Strict label error rate (REL = 1) bounded at only **3.4%** [95% CI: 2.3%--5.1%]
> > > - All scores significantly above 3.0 (Wilcoxon $p < 0.001$; rank-biserial $r = 0.714$; Cohen's $d = 0.734$)
> > >
> > > **Table R1.** Per-category results (H₁: REL > 3.0, 100 samples each).
> > >
> > > | Category | Description | Mean REL | Acceptable ($\geq$ 3) | Cohen's $d$ | Rank-biserial $r$ |
> > > |:---------|:------------|:--------:|:-----:|:-----:|:-----:|
> > > | 00A | Sound Effects Only | 3.45 | 74.0% | 0.365 | 0.422 |
> > > | 0M0 | Music Only | 3.77 | 80.0% | 0.644 | 0.648 |
> > > | 0MA | Music + Sound Effects | 3.57 | 79.0% | 0.453 | 0.511 |
> > > | S00 | Speech Only | 4.10 | 95.0% | 1.247 | 0.935 |
> > > | S0A | Speech + Sound Effects | 3.89 | 88.0% | 0.838 | 0.788 |
> > > | SM0 | Speech + Music | 4.07 | 94.0% | 1.250 | 0.915 |
> > > | SMA | Speech + Music + Sound | 3.85 | 87.0% | 0.790 | 0.744 |
> > >
> > > > All 7 categories pass Wilcoxon test ($p < 0.001$) after Bonferroni correction.
> > >
> > > The **measured severe error rate is only 3.4%**, confirming the multi-expert pipeline effectively prevents cascading errors.
> > >
> > > ---
> > >
> > > ### 1.5 New Evidence: DATE Bootstrap Stability
> > >
> > > Our initial rebuttal showed $n = 150$ is over-powered (Cohen's $d = 0.89$, power > 99.99%). We now add **complementary evidence**: a bootstrap stability test using Gemini-3.0-Pro across **9 evaluation groups** (Speech, Music, Sound each split into Pure/Mixed with independent reference pools, plus Long, Short, Environment), 50 iterations per ratio, **each fully re-computing DATE including discrimination-rank**.
> > >
> > > **Table R2.** Full-data (100%) bootstrap summary.
> > >
> > > | Group | Size | DATE Mean | CV (%) | 95% CI | Max Dev |
> > > |:------|-----:|----------:|-------:|:-------|:--------|
> > > | Long | 20,052 | 0.6486 | 0.14 | [0.647, 0.650] | $\pm$0.002 |
> > > | Short | 20,052 | 0.6581 | 0.16 | [0.656, 0.660] | $\pm$0.003 |
> > > | Pure-Speech | 7,839 | 0.6049 | 0.29 | [0.602, 0.607] | $\pm$0.004 |
> > > | Mixed-Speech | 8,593 | 0.6236 | 0.31 | [0.620, 0.627] | $\pm$0.004 |
> > > | Pure-Music | 2,593 | 0.4967 | 0.77 | [0.490, 0.503] | $\pm$0.008 |
> > > | Mixed-Music | 8,593 | 0.3988 | 0.46 | [0.395, 0.402] | $\pm$0.004 |
> > > | Pure-Sound | 848 | 0.5520 | 1.04 | [0.542, 0.565] | $\pm$0.014 |
> > > | Mixed-Sound | 8,593 | 0.2992 | 0.94 | [0.294, 0.304] | $\pm$0.007 |
> > > | Environment | 20,052 | 0.2613 | 0.58 | [0.259, 0.264] | $\pm$0.003 |
> > >
> > > **Table R3.** CV(%) across subset sizes. CV@$x$% = CV from 50 bootstrap iterations each drawing $x$% of full data with replacement.
> > >
> > > | Group | CV@1% | CV@5% | CV@10% | CV@20% | CV@50% | CV@100% |
> > > |:------|------:|------:|-------:|-------:|-------:|--------:|
> > > | Long | 1.79 | 0.61 | 0.54 | 0.32 | 0.22 | 0.14 |
> > > | Short | 1.60 | 0.58 | 0.55 | 0.36 | 0.22 | 0.16 |
> > > | Pure-Speech | 3.14 | 1.37 | 0.92 | 0.68 | 0.40 | 0.29 |
> > > | Mixed-Speech | 3.53 | 1.22 | 0.88 | 0.67 | 0.47 | 0.31 |
> > > | Pure-Music | 10.65 | 4.10 | 3.22 | 2.29 | 1.32 | 0.77 |
> > > | Mixed-Music | 5.86 | 2.69 | 2.28 | 1.37 | 0.67 | 0.46 |
> > > | Pure-Sound | 18.62 | 5.85 | 4.61 | 3.22 | 1.68 | 1.04 |
> > > | Mixed-Sound | 9.60 | 4.36 | 3.11 | 2.77 | 1.28 | 0.94 |
> > > | Environment | 5.60 | 2.92 | 1.90 | 1.32 | 0.98 | 0.58 |
> > >
> > >  8/9 groups exhibit CV < 1% at full scale; only Pure-Sound (1.04%, N=848) slightly exceeds 1%. CV drops below 3% at 5% sampling for 7/9 groups. Combined with the 150-pair human validation (power > 99.99%, concordance 84%), DATE's reliability is confirmed from both human-alignment and metric-stability perspectives.
> > >
> > > ---
> > >
> > > ### Summary
> > >
> > > -  **700-sample human audit** (20 experts) -- severe error rate 3.4%, all 7 categories significantly above baseline
> > >
> > > -  **Bootstrap stability** (9 groups, up to 20,052 samples) -- CV < 1% at full scale for 8/9 groups

---

### Decision · Program_Chairs · 2026-04-30

**Decision:**

Accept (regular)

**Comment:**

Although one reviewer maintained a principled objection regarding the standard of per-sample verification expected for benchmarks, this concern remained a minority view. Following the authors’ detailed rebuttal and new quantitative validation, other reviewers revised their assessments and converged on a positive evaluation. Based on the updated final justifications and overall reviewer consensus, I recommend acceptance.